# Human spinal cord in vitro differentiation pace is initially maintained in heterologous embryonic environments

Alwyn Dady*, Lindsay Davidson, Pamela A Halley, Kate G Storey*

Division of Cell and Developmental Biology, School of Life Sciences, University of Dundee, Dundee, United Kingdom

**Abstract** Species-specific differentiation pace in vitro indicates that some aspects of neural differentiation are governed by cell intrinsic properties. Here we describe a novel in vitro human neural-rosette assay that recapitulates dorsal spinal cord differentiation but proceeds more rapidly than in the human embryo, suggesting that it lacks endogenous signalling dynamics. To test whether in vitro conditions represent an intrinsic differentiation pace, human iPSC-derived neural rosettes were challenged by grafting into the faster differentiating chicken embryonic neural tube iso-chronically, or hetero-chronically into older embryos. In both contexts in vitro differentiation pace was initially unchanged, while long-term analysis revealed iso-chronic slowed and hetero-chronic conditions promoted human neural differentiation. Moreover, hetero-chronic conditions did not alter the human neural differentiation programme, which progressed to neurogenesis, while the host embryo advanced into gliogenesis. This study demonstrates that intrinsic properties limit human differentiation pace, and that timely extrinsic signals are required for progression through an intrinsic human neural differentiation programme.

## Editor's evaluation

This manuscript is of interest to a large audience in the fields of stem cells, developmental biology and neural regeneration. The authors assess the roles of extrinsic versus intrinsic signalling on differentiation of human neural cells by comparing their differentiation rates across different environments (in vitro, in the human embryo and grafted into a chicken embryo).

*For correspondence:
a.dady@dundee.ac.uk (AD);
k.g.storey@dundee.ac.uk (KGS)

Competing interest: The authors declare that no competing interests exist.

## Introduction

Understanding human embryonic development is a major challenge in contemporary biology and is in large part investigated by creation of pluripotent cell derived in vitro models of developing tissues and organs. One of the most striking features of such in vitro assays is the remarkably faithful recapitulation of specific differentiation programmes in minimal culture conditions. For example, species specific patterns of cell proliferation and timing of neuronal sub-type differentiation are evident in human and macaque cortical progenitors, even when these cells are cultured together (*Otani et al., 2016*), strongly supporting the notion that cell autonomous mechanisms direct differentiation programmes. Recent analyses of potential mechanisms that account for human specific differentiation tempo have identified general protein stability and cell cycle duration as parameters that constrain this process in spinal cord progenitors (*Rayon et al., 2020*). Moreover, a human specific gene, *ARHGAP11B*, has recently been shown to promote the enhanced proliferative capacity of human basal cortical progenitors by augmenting a specific metabolic pathway (*Namba et al., 2020*); and so uncovering a genetic basis for such cell autonomous cell behaviour. Such findings indicate that cell intrinsic mechanisms

underpin differentiation programme progression and suggest that this process can proceed from the neural progenitor cell state with minimal extrinsic signals. Evaluation of the extent to which human neural progenitors differentiate cell autonomously requires a detailed understanding of the differentiation timing of specific cell types during normal human embryogenesis and how well this is recapitulated in vitro. Indeed, comparison of the timing of the appearance of specific cell types in the human embryo with in vitro data may also help to inform better modelling approaches, which largely rely on morphological changes to align differentiation tempo (*Workman et al., 2013*; *Otani et al., 2016*).

In other vertebrate embryos, the extracellular environment, including spatially and temporally regulated signals provided by neighbouring tissues, is known to orchestrate neural differentiation and such signalling is likely to direct this process in human embryos. In vitro models, from rosettes to more complex three-dimensional organoids and gastruloids, aim to recreate this by provision of supportive extracellular environments and timely exposure to ligands or small molecules that modulate signalling activity (*Chambers et al., 2009*; *Lancaster et al., 2013*; *Otani et al., 2016*; *Ogura et al., 2018*; *Chiaradia and Lancaster, 2020*; *Moris et al., 2020*). A critical test of the extent to which neural progenitor differentiation is governed by cell autonomous mechanisms is their behaviour in a heterologous environment. However, studies examining differentiation pace and accuracy in such contexts are lacking. This is obviously important when considering use of such neural progenitors in cell replacement strategies following injury or disease (*Sofroniew, 2018*) or for local delivery of secreted factors for therapeutic effect (*Zhang et al., 2017*), as these approaches involve transplantation into heterologous environments and yet anticipate differentiation into specific cell types (e.g., *Kumamaru et al., 2018*; *Kumamaru et al., 2019*).

A relatively simple region of the developing nervous system in which to evaluate the extent to which human neural progenitors differentiate cell autonomously, is the developing spinal cord. In vitro generation of spinal cord progenitors from mammalian pluripotent cells has been achieved by approaches in which anterior neural tissue is induced and then caudalised (e.g., *Meinhardt et al., 2014*; *Gupta et al., 2018*). Recent advances in understanding the cellular origins of the spinal cord have further shown that it arises from a bipotent epiblast cell population adjacent to the anterior primitive streak known as neuromesodermal progenitors (NMPs) (*Tzouanacou et al., 2009*; *Gouti et al., 2014*; *Gouti et al., 2015*; *Henrique et al., 2015*; *Gouti et al., 2017*; *Steventon and Martinez Arias, 2017*; *Binagui-Casas et al., 2021*). Moreover, in vitro generation of spinal cord progenitors from NMPs may align better with the temporal sequence of cell fate decisions in the embryo. Derivation of NMP-like cells from pluripotent cells has been demonstrated (*Turner et al., 2014*; *Gouti et al., 2017*; *Frith et al., 2018*; *Verrier et al., 2018*; *Frith and Tsakiridis, 2019*; *Edri et al., 2019*) and this has catalysed investigation of mechanisms regulating human as well as mouse spinal cord development.

Here, we present a novel human pluripotent cell-derived in vitro neural rosette assay that recapitulates dorsal spinal cord differentiation observed in vertebrate embryos and use this to evaluate the extent to which cell intrinsic properties direct differentiation of human neural progenitors. Using key cell type-specific marker proteins, we compare in detail in vitro neural differentiation with that in the human embryonic spinal cord. In a series of grafting experiments, in which human iPSC derived spinal cord progenitors are transplanted into the chicken embryonic neural tube, we further test whether differentiation pace can be increased in this faster differentiating environment, as well as the extent to which the differentiation trajectory of transplanted cells relies on timely provision of extrinsic signalling.

## Results

### In vitro generation of human dorsal spinal cord rosettes

An in vitro protocol which generates dorsal spinal cord from human pluripotent cell-derived NMP-like cells with brachial/ thoracic character (*Verrier et al., 2018*) was used to make neural rosettes (*Figure 1A*). The precise timing with which NMP-like cells (derived from H9 human Embryonic Stem Cells - ESCs) enter the neural differentiation programme was assessed by quantifying cells expressing a range of neural progenitor markers during the first 10 days of two-dimensional culture in a minimal medium (*Figure 1A, B*, *Figure 1—source data 1* and *Figure 1—source data 2*). This early neural tissue formed compact condensations of cells by day 4 (D4) of NMP-like cell differentiation, which lacked overt apico-basal polarity as indicated by localisation of N-Cadherin (Cadherin-2) at most cell-cell interfaces (*Hatta and Takeichi, 1986*; *Dady et al., 2012*; *Figure 1B*). These cells expressed

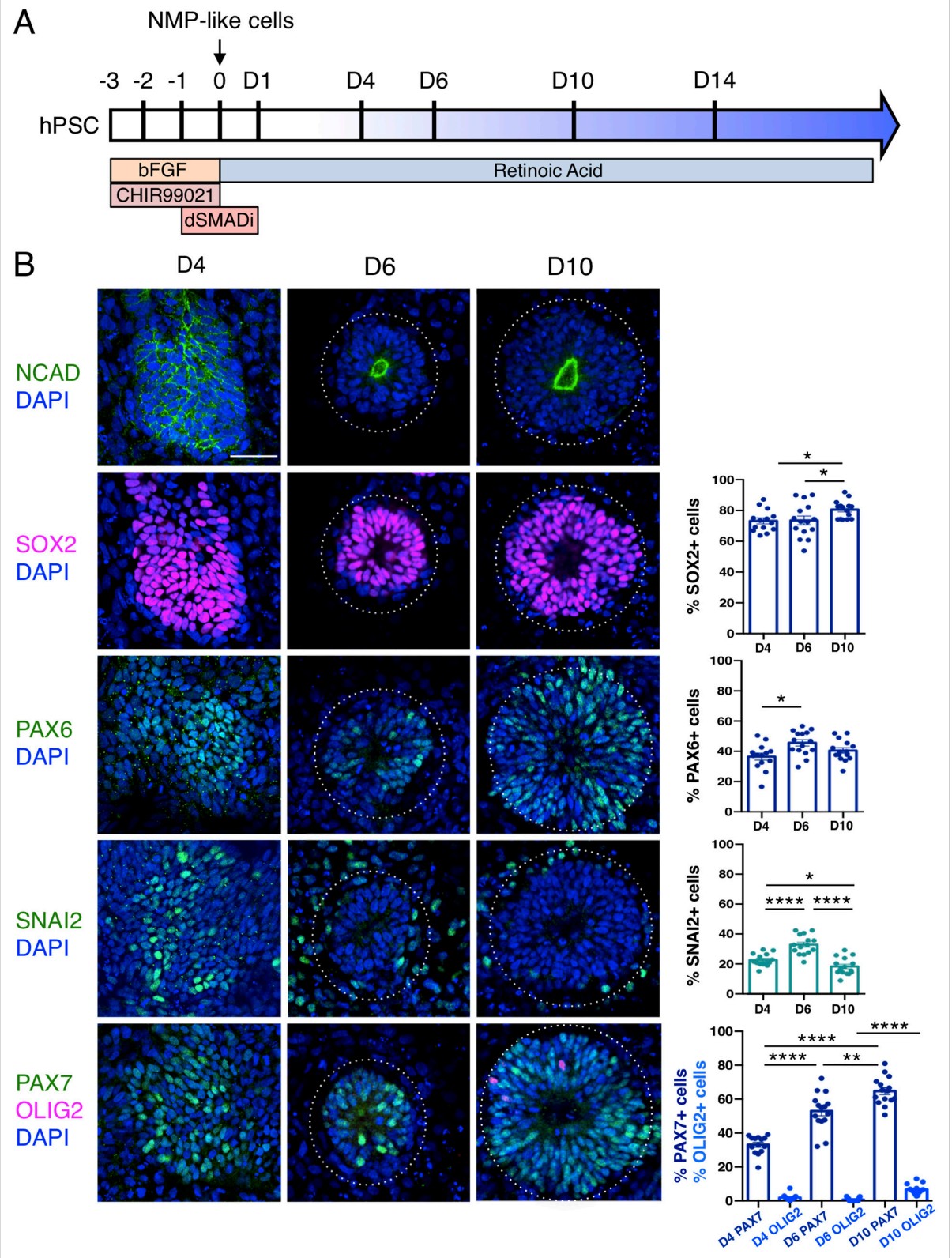

**Figure 1.** Human pluripotent cell derivation of NMP-like cells and their differentiation into dorsal spinal cord neural rosettes. (**A**) Timeline and protocol for induction and differentiation of dorsal spinal cord neural rosette from human pluripotent stem cells (hPSC); (**B**) Expression patterns and quantification of cell type specific proteins in rosettes on days D4, D6, and D10 detected by immunofluorescence (IF) combined with DNA stain DAPI: apical polarity marker N-Cadherin (NCAD), neural progenitor markers SOX2, PAX6, early neural crest marker SNAIL2, and dorso-ventrally restricted neural

*Figure 1 continued on next page*

Figure 1 continued

progenitor markers PAX7 and OLIG2. Quantifications indicate proportions of expressing cells in either a defined field on D4 or within a rosette from D6, in 15 samples for each marker (*n* = 5 rosettes from each of 3 independent differentiations, each data point represents a single rosette), see section 'Materials and Methods'. Data analysed with Mann–Whitney test, errors bars ± SEM, p values: *p<0.05; **p<0.01; **** p<0.0001; scale bar = 50 μm.

The online version of this article includes the following source data for figure 1:

**Source data 1.** Quantifications indicating proportions of cells expressing SOX2, PAX6 and SNAI2 in either a defined field on D4 or within a rosette from D6 and D10.

**Source data 2.** Quantifications indicating proportions of cells expressing PAX7 and OLIG2 in either a defined field on D4 or within a rosette from D6 and D10.

**Source data 3.** Quantifications indicating the different diameters of rosette in micrometre from D6, D10, and D14.

SOX2, characteristic of both pluripotent, NMP and neural progenitor cells (*Uchikawa et al., 2003*; *Uchikawa et al., 2011*; *Bylund et al., 2003*; *Ellis et al., 2004*), and the neural progenitor protein PAX6 (*Walther and Gruss, 1991*; *Pevny et al., 1998*). Some cells expressing the early neural crest marker SNAI2 were also detected on D4 (*Nieto, 2002*; *Jiang et al., 2009*; *Dady and Duband, 2017*; *Figure 1B*, *Figure 1—source data 1*). The expression of PAX7, which distinguishes dorsal neural progenitors (*Jostes et al., 1990*; *Gruss and Walther, 1992*) and the absence of OLIG2, characteristic of more ventral spinal cord (*Mizuguchi et al., 2001*; *Novitch et al., 2001*) further indicated that at D4 this differentiation protocol had generated human neural tissue with dorsal spinal cord identity (*Figure 1B*, *Figure 1—source data 2*).

By D6 neural rosettes composed of apico-basally polarised SOX2, PAX6, and PAX7 expressing progenitors had formed, with a single, central lumen and peripheral SNAI2 neural crest cells (*Figure 1B*). This spatial reorganisation was accompanied by decline in SNAI2+ cells and the appearance, by D10, of a scattering of OLIG2 expressing cells (*Figure 1B*, *Figure 1—source data 1*). This emergence of neural rosettes from a condensing epiblast cell population at D4, may equate to the formation of the neural plate and establishment of the pseudostratified neuroepithelium, but it is also reminiscent of the phases of embryonic secondary neurulation (*Schoenwolf and Nichols, 1984*; *Catala et al., 1995*; *Dady et al., 2012*; *Dady et al., 2014*; *Fedorova et al., 2019*).

These findings show that differentiation of human ESC derived NMP-like cells in a minimal culture medium containing retinoic acid (RA) is sufficient to generate self-organising dorsal spinal cord rosettes and accompanying neural crest.

## Sequential differentiation of human dorsal spinal cord progenitors is recapitulated in vitro

Dorsal spinal cord rosettes were further characterised as they progressed through their differentiation programmes (*Figure 2A*). Decline of SNAI2-expressing cells at D10 coincided with the emergence of migrating neural crest cells expressing HNK1 (*Tucker et al., 1984*; *Bronner-Fraser, 1986*), SOX10 (*Bondurand et al., 1998*), and TFAP2α (*Zhang et al., 1996*; *Betters et al., 2010*) in cells now just outside the rosettes (*Jiang et al., 2009*; *Figure 2B*, *Figure 2—source data 1*). The onset of neuron production was assessed in several ways: (i) by analysis of immature neuronal marker Doublecortin (DCX) expression (*Gleeson et al., 1999*; *Brown et al., 2003*) (using a hESC [H1] neuronal reporter cell line in which a yellow fluorescent protein [*YFP*] sequence is fused to endogenous *DCX* [*Yao et al., 2017*; *Figure 2C*, *Figure 2—source data 2*]),and (ii) by characterisation of cdk inhibitor P27/KIP1 expression, which identifies post-mitotic cells in the neural tube (*Gui et al., 2007*; *Figure 2D*, *Figure 2—source data 3*). These analyses defined a neurogenesis timeline which commenced by D8, peaked at D14 and progressed through to D20.

By D20 DCX-expressing neurons largely resided at the rosette perimeter and the early glial progenitor marker, the transcription factor Nuclear factor 1 A-type (NFIA) (*Deneen et al., 2006*), was now detected in cells within the rosette (*Figure 2C*, *Figure 2—source data 2*). This indicates operation of a switch from neurogenic to gliogenic differentiation programmes in the human spinal cord rosettes that is similar to that observed in the mouse embryo (*Deneen et al., 2006*).

Finally, we determined the identity of neurons generated in this in vitro assay by immunofluorescence of dorsal interneuron (dIs) subtype-specific markers. In the mouse, embryonic neural tube distinct sensory interneurons arise in specific dorsoventral positions and express distinguishing marker

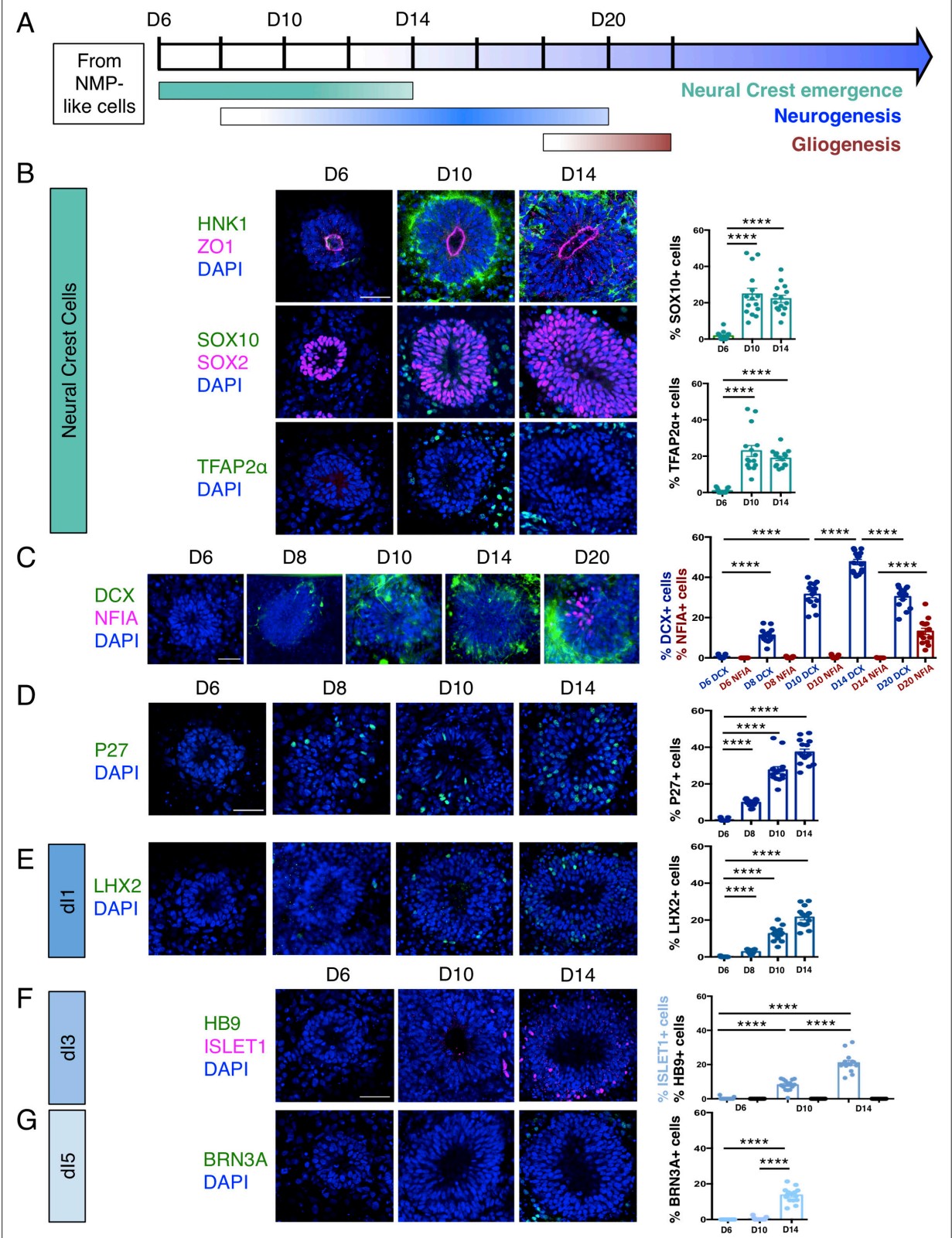

**Figure 2.** Human dorsal spinal cord rosettes exhibit stereotyped sequence of cell-type specific differentiation. (**A**) Schematic of in vitro sequence of cell-type specific differentiation of dorsal spinal cord rosettes; (**B**) emergence of migrating neural crest from rosettes by D10, documented by IF for key marker proteins HNK1, (lumen defined by apical marker ZO-1), SOX10 (neural progenitors defined by SOX2), and TFAP2α and quantified for SOX10 and TFAP2α; (**C**) transition from neurogenesis to gliogenesis documented and quantified in immature neuronal reporter line (H1:DCX-YFP) with co-IF for

*Figure 2 continued on next page*

*Figure 2 continued*

gliogenesis marker Nuclear Factor I-A (NFIA) at key timepoints; (**D**) P27/KIP1 expression which by analogy identifies most dorsal post-mitotic neurons, confirms neurogenesis onset by D8; (**E–G**) emergence of dorsal interneuron (dIs) subtypes by D10 indicated by dI1s/LHX2 (**E**), dI3s/ISLET1 (**F**) (note lack of HB9/ ISLET1 co-expression distinguishes these cells from motor neurons) and dI5s/BRN3A. Sample numbers, statistics and error bars as in *Figure 1*, each data point represents a single rosette and see section 'Materials and Methods'; scale bar = 50 μm.

The online version of this article includes the following source data and figure supplement(s) for figure 2:

**Source data 1.** Quantifications indicating proportions of cells expressing migratory Neural Crest Cells markers (SOX10 and TFAP2α) within a rosette from D6, D10, and D14.

**Source data 2.** Quantifications indicating proportions of cells expressing neuronal marker (DCX) and glial marker (NFIA) within a rosette from D6, D8, D10, D14, and D20.

**Source data 3.** Quantifications indicating proportions of cells expressing neuronal marker (P27) within a rosette from D6, D8, D10 and D14.

**Source data 4.** Quantifications indicating proportions of cells expressing dI1s marker (LHX2) within a rosette from D6, D8, D10 and D14.

**Source data 5.** Quantifications indicating proportions of cells expressing ISLET1 and HB9 within a rosette from D6, D10, and D14.

**Source data 6.** Quantifications indicating proportions of cells expressing dI5s marker (BRN3A) within a rosette from D6, D8, D10, and D14.

**Figure supplement 1.** Dorsal interneurons type two expressing LHX1, dI2/LHX1, are not generated in human spinal cord neural rosettes in vitro.

---

proteins (*Lai et al., 2016*). By D8, expression of LHX2 indicative of proprioceptive interneurons (dI1s) was first detected (*Figure 2E*, *Figure 2—source data 4*) and accompanied by D10 by ISLET1 positive mechanosensory interneurons (dI3s) (*Figure 2F*, *Figure 2—source data 5*; *Liem et al., 1997*). By D14, dI5 interneurons expressing BRN3A were detected (*Ninkina et al., 1993*; *Fedtsova and Turner, 1995*; *Figure 2G*, *Figure 2—source data 6*). Moreover, ISLET1+ cells did not co-express the transcription factor HB9, a marker of spinal cord motoneurons (*Arber et al., 1999*), confirming their dI3 identity (*Figure 2F*). In contrast, LHX1/dI2s/dI4s expressing cells were not identified (*Figure 2—figure supplement 1*), indicating that this protocol is insufficient to generate all types of dorsal interneurons.

These data document the differentiation capacity in this human spinal cord rosette assay and demonstrate that it reproducibly generates dorsal neural differentiation programmes. These begin with dorsal neural progenitors and emerging neural crest, progress to neurogenesis (of specific dorsal interneurons dI1s, dI3s, and dI5s) and a later switch to gliogenesis: a sequence that recapitulates the temporal order of differentiation observed in the spinal cord of mouse and avian embryos (*Figure 2A*; *Deneen et al., 2006*; *Glasgow et al., 2017*; *Andrews et al., 2017*).

## Human spinal cord differentiation in vitro progresses more rapidly than in the human embryonic spinal cord

This conservation in the sequence of the dorsal neural differentiation programme contrasts with the longer duration of human embryogenesis in comparison with that observed in the mouse and chicken. Indeed, the pace of differentiation in the developing nervous system and that of body axis segmentation is slower in human embryos (*Rayon et al., 2020*; *Matsuda et al., 2020*). These latter studies indicate the presence of species-specific cell intrinsic differences that govern developmental tempo. To understand how well this tempo is captured in our in vitro assay, we next sought to compare the sequence and timing of differentiation progression in human spinal cord rosettes with that observed in the human embryo.

The pattern and timing of key marker protein expression in human embryonic spinal cord was investigated using tissue samples provided by the HDBR (*Figure 3A–H* and *Figure 3—figure supplement 1*) and available data. One way in which to assess differentiation progression is to take advantage of the spatial separation of the temporal events of differentiation along the embryonic body axis, and so analyses were carried out at distinct rostro-caudal levels in Carnegie stage (CS) 12/13 (gestational weeks 4–4.5) human embryonic spinal cord (sections from two embryos were analysed, one at CS12 and one at CS13, both showed the same expression patterns) *O'Rahilly, 1987*; sacral (the most caudal and least differentiated region), lumbar and thoracic/brachial (more rostral and progressively more differentiated regions). Developmental time along the body axis can be approximated by that indicated by segmentation progression from the sacral to thoracic/brachial region, which involves generation of ~20 somites with one human somite generated every 5 hr (*Turnpenny et al., 2007*; *Hubaud and Pourquié, 2014*; *Matsuda et al., 2020*) this represents, 100 hr or ~4 days (*Figure 3A*).

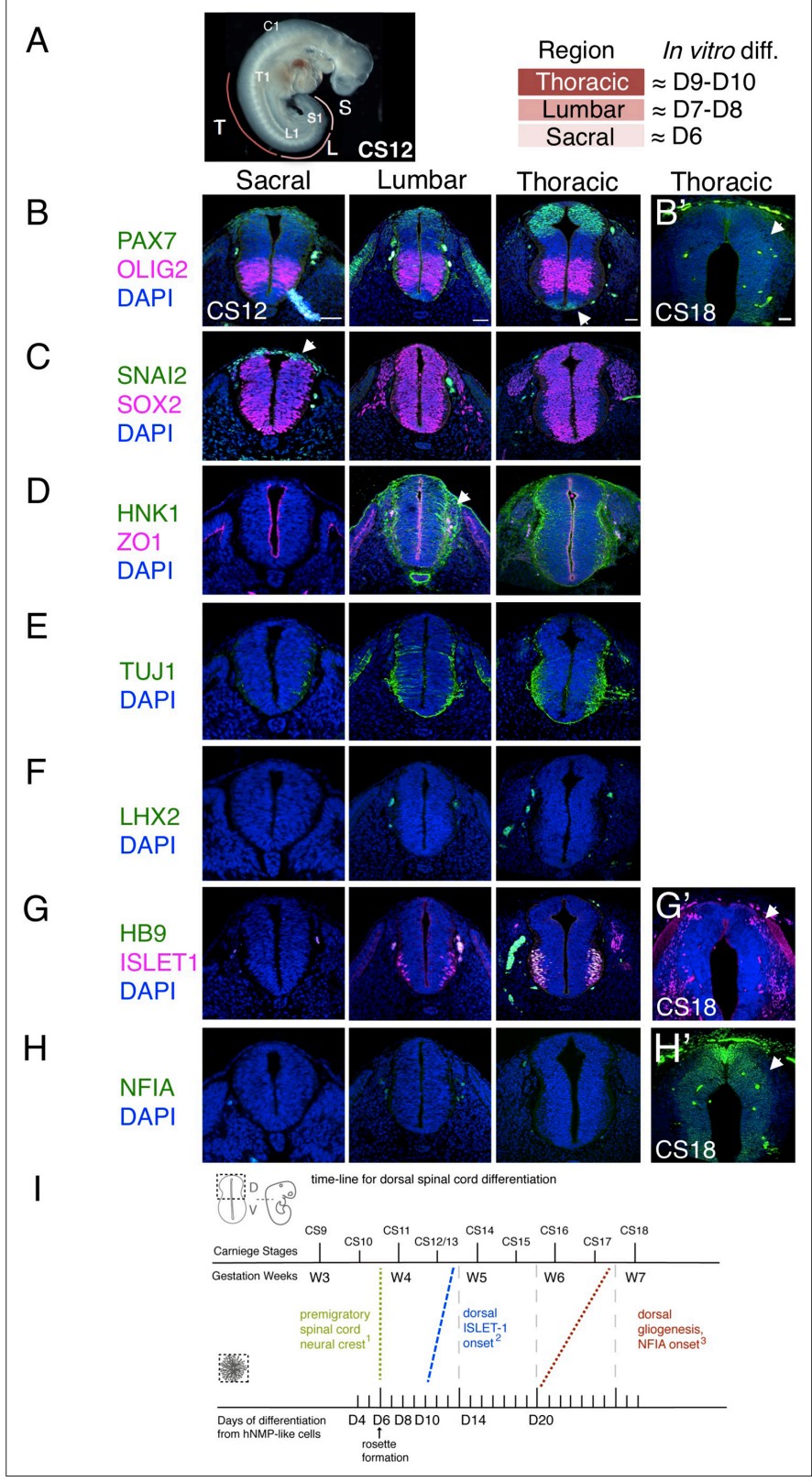

**Figure 3.** Characterization of neural progenitors and neuronal subtypes in human embryonic spinal cord and comparison with in vitro human neural rosette differentiation. (**A**) CS12 human embryo with anatomical sub-regions indicated and their alignment with in vitro spinal cord rosette differentiation (see text); (**B–H**) transverse sections through human embryonic spinal cord at sacral, lumbar and thoracic levels at 4/4.5 weeks (CS12/CS13) and at

*Figure 3 continued on next page*

*Figure 3 continued*

brachial/thoracic at 7/7.5 weeks (CS18) analysed by IF for expression of (**B, B'**) dorsal, PAX7, or ventral, OLIG2, spinal cord progenitor markers (note novel PAX7 detection in the floor plate, arrow) and (**B'**) waning PAX7 at CS18 (arrow); (**C**) premigratory neural crest cell marker SNAI2 is only detected in sacral regions (arrow) shown here with pan-neural progenitor marker SOX2; (**D**) differentiating migrating neural crest expressing HNK1 emerges from the lumbar region (arrow); (**E**) neuronal differentiation detected with pan-neuronal marker TUJ1 is manifest dorsally in lumbar regions; dorsal interneurons indicated by (**F**) dl1 specific transcription factor LHX2 and (**G**) dl3 identifying ISLET1 expressing cells were not detected dorsally in any region of the spinal cord at CS12/CS13; (**G'**) dorsal ISLET1 interneurons at CS18; (**H, H'**) gliogenesis marker NFIA detected dorsally only at CS18; (**I**) Schematic aligning dorsal brachial/thoracic spinal cord differentiation in human embryo and in hPSC derived dorsal spinal cord rosettes, based on (1) *Bondurand et al., 1998*, *O'Rahilly and Müller, 2007*; (2) This study, *Rayon et al., 2020*, *Marklund et al., 2014*; (3) This study, *Rayon et al., 2020*; three sections were analysed at each level for each marker in $n$ = 2 CS12/13 and $n$ = 1 CS18 human embryos. Scale bar = 50 μm.

The online version of this article includes the following figure supplement(s) for figure 3:

**Figure supplement 1.** Expression patterns of neuronal markers P27 and ISLET1 in human embryonic spinal cord.

The sacral spinal cord at CS12/13 stage was composed of progenitors expressing dorsal (PAX7) and ventral (OLIG2) markers (*Figure 3B*) as well as pre-migratory neural crest cells expressing SNAI2 (*Figure 3C*) and lacking migratory neural crest cell marker HNK1 (*Figure 3D*). Moreover, neurogenesis, revealed by the pan-neuronal marker TUJ1, was only detected ventrally (*Menezes and Luskin, 1994*; *Figure 3E*). The differentiation state in the dorsal region of the CS12/13 sacral neural tube therefore aligns well with that found in D6 dorsal spinal cord rosettes, which constitute apico-basally polarised neuroepithelium expressing PAX6, PAX7, and SNAI2 (*Figure 1B*) that lacks migratory neural crest and neurons (*Figure 2B–G*).

In the next more rostral, lumbar region, PAX7 and OLIG2 domains have enlarged (*Marklund et al., 2014*), SNAI2-expression is no longer detected (*Figure 3C*), and migrating HNK1-expressing neural crest cells have now emerged (*Figure 3D*). The first neurons have now appeared in the dorsal half of the neural tube as indicated by TUJ1 expression, but neuronal subtype markers LHX2, LHX1, dorsal ISLET-1 have yet to be detected, with co-expression of ISLET1 and HB9 identifying ventrally located motor-neurons (*Arber et al., 1999*; *Rayon et al., 2020*), (*Figure 3F and G* and *Figure 2—figure supplement 1*). This combination of marker proteins indicates that the lumbar region broadly corresponds to D8 neural rosettes in which the first neurons have formed, as indicated by DCX and P27 expression (*Figure 2C, D*); detection of HNK1 in the lumbar region supports this alignment, as this is not found in D6 rosettes and is robustly expressed by D10. The identity of these first dorsal neurons is not yet clear, we detect LHX2 expressing cells in D8 rosettes, but neither LHX1 nor ISLET-1 positive cells were found at this time (*Figure 2E, F* and *Figure 2—figure supplement 1*). Given the alteration in transcription factor combinatorial code created by the expansion of the OLIG2 domain in human spinal cord progenitors (*Marklund et al., 2014*), one possibility is that human-specific neuronal cell types arise in this region.

Importantly, dorsal interneuron markers LHX2 (dl1) and ISLET1/HB9 (dl3) were also not detected at thoracic levels (which included sections from presumptive forelimb level, brachial spinal cord), (*Figure 3F, G*). This is consistent with the lack of dorsal ISLET1 expression at CS13 at brachial levels in a recent report (*Figure 1C*; *Rayon et al., 2020*), although a few dorsal ISLET1 positive cells were found in brachial spinal cord of another CS13 embryo (*Figure 3—figure supplement 1* and T. Rayon pers comm.). This suggests that these interneurons are just beginning to be generated at CS13 at thoracic/ brachial levels and is consistent with detection of LHX2 expressing neurons in CS14 spinal cord (*Marklund et al., 2014*) and (see *Rayon et al., 2020*; *Rayon et al., 2021*). In contrast, LHX2 positive cells were first detected in D8 rosettes and both LHX2 and ISLET1 positive cells were found in all D10 neural rosettes (*Figure 2E, F*). This earlier detection of dorsal neurons in rosettes suggests that the differentiation programme progresses more rapidly in vitro than in the human embryonic spinal cord (*Figure 2I*).

A further way to assess differentiation tempo is to align progression of a defined cell population in vitro and in vivo using time elapsed from a key landmark event. The NMP-like cells generated in our protocol have a brachial/thoracic regional character as indicated by Hox gene expression (*Verrier et al., 2018*) and differentiation of this region can be compared in vitro and in embryos over time.

Detailed analysis of the first appearance of spinal cord neural crest in the human embryo provides a clear landmark with which to align in vivo and in vitro differentiation (*Bondurand et al., 1998*; *O'Rahilly and Müller, 2007*). These latter studies identify morphological emergence of migrating neural crest in cervical level spinal cord in the CS11 human embryo (*O'Rahilly and Müller, 2007*) and in even more caudal spinal cord at this stage using *SOX10* mRNA expression (*Bondurand et al., 1998*). From this we infer that pre-migratory neural crest is present prior to CS11 in brachial/thoracic spinal cord, and so aligns this region in CS10-11 embryos with D6 in our neural rosettes, which express SNAI2 but lack migratory neural crest marker proteins SOX10, HNK1, and TFAP2α (*Figures 1B and 2B*). Using this alignment point we then compared time to appearance of later cell type specific markers in neural rosettes and human embryos (*Figure 3I*). As above, dorsally located ISLET1/dI3s first appear ~CS13 (*Figure 3—figure supplement 1*) and are robustly detected at CS15 (*Rayon et al., 2020*; *Rayon et al., 2021*; and see HDBR Atlas http://hdbratlas.org), and here this indicates that these neurons appear ~4 days earlier in D10 rosettes (*Figures 2E, F and 3I*). The landmark switch from neurogenesis to gliogenesis indicated by NFIA expression in dorsal neural progenitors was detected in human embryos at CS18 (gestational weeks 7–7.5) (*Figure 3H'*) but not at CS17 (*Rayon et al., 2020*). This is ~1 week later than in neural rosettes, where NFIA was robustly found at D20 (*Figures 2C and 3I*). Moreover, the overall time from spinal cord migratory neural crest appearance to dorsal gliogenesis takes ~2 weeks (D6 –D20) in human dorsal spinal cord rosettes in vitro and ~3–3.5 weeks (CS10/11-CS17/18) in the human embryonic spinal cord (*Figure 3I*). These further comparisons also suggest that differentiation of neural rosettes in vitro proceeds more quickly than equivalent neural progenitor cells in vivo.

## In vitro differentiation pace of human neural progenitors is initially unaltered by iso-chronic grafting into the more rapidly developing chicken embryonic environment

In the vertebrate embryo, the timing of neural differentiation is orchestrated by signals from adjacent tissues, which drive and coordinate axial differentiation but the localised and temporally controlled characteristics of this regulatory influence are lacking in vitro. Indeed, in the embryo, neural progenitors will generate more cells to make a bigger structure and so proliferative phases that expand cell populations might be longer. We have found that the smaller rosette structures generated in vitro differentiate more rapidly and this raises the possibility that differentiation pace in this minimal assay reflects species-specific intrinsic constraints on this process (*Rayon et al., 2020*). To test whether rosette cells retain in vitro differentiation timing or if this can run still faster in a more rapidly developing embryo that also provides a more coherent extracellular environment, we next iso-chronically grafted human neural rosettes into the dorsal neural tube of the chicken embryo.

To this end, human induced pluripotent stem cells (iPSCs) (Cellartis hIPS4 cell line engineered to express a plasma membrane tagged GFP, ChiPS4-pmGFP, see section 'Materials and Methods') were differentiated into NMP-like cells and then into D6 neural rosettes as described above (*Figure 1A*). Multiple plates of rosettes were generated so that control in vitro differentiation of iPSC derived rosettes was monitored in parallel with that of D6 rosettes grafted into the chicken embryo. The expression of key marker proteins in these ChiPS4-pmGFP derived rosettes was found to be the same as that in D6 rosettes derived from the hESC H9 line (*Figure 4—figure supplement 1*, *Figure 4—figure supplement 1—source data 1*). Individual neural rosettes were micro-dissected, lightly dissociated and grafted in place of one side of the dorsal neural tube, adjacent to newly formed somites in stage HH10-11 (Embryonic day 2, E2) chicken embryos (*Figure 4A–B'*, *Figure 4—figure supplement 2*).

The precision of dorsal neural tube removal was evaluated in a subset of embryos immediately after operation and this showed that tissue removed included only neural crest and neural progenitors expressing respectively SNAI2 and PAX7, as found in D6 human neural rosettes (*Figure 4—figure supplements 1 and 2*). To control for any disturbance in the timing of differentiation due to transplantation procedures we monitored expression of key marker proteins following transplantation of dorsal neural tube from transgenic myr-GFP-expressing chicken embryos at the same stage and into the same position in wildtype chicken embryos. This demonstrated the robust ability of the E2 chicken embryo to accommodate the grafting procedure and no differences were found between transplanted and the contralateral un-operated side of the neural tube after 2 days (*Figure 4—figure supplement*

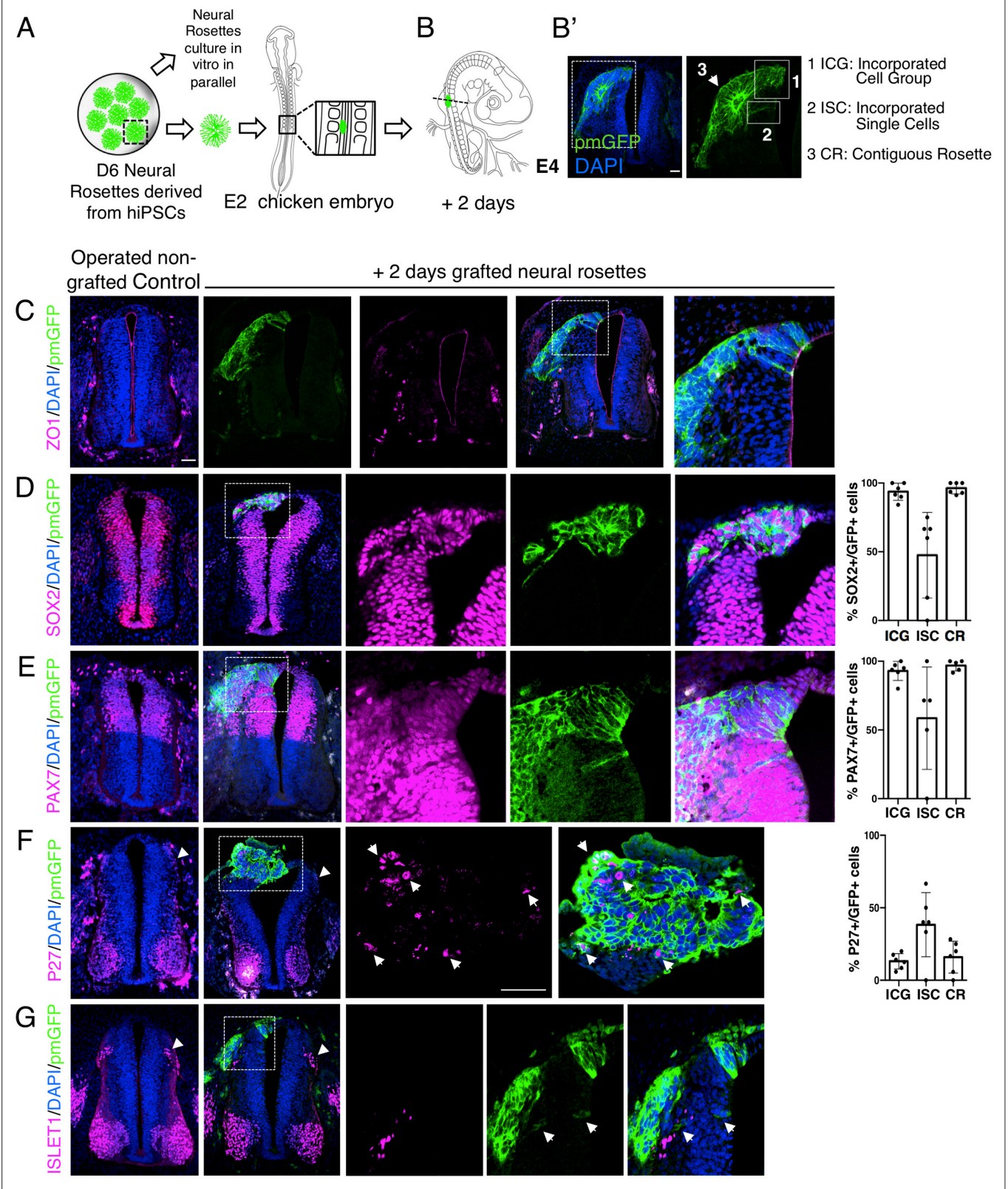

**Figure 4.** Iso-chronically transplanted hiPSC-derived neural rosettes exhibit cell intrinsic differentiation despite exposure to more rapidly developing chicken embryonic environment. (**A**) Schematic of human dorsal spinal cord neural rosette transplantation experiment; (**B-B′**) two days after grafting human cells were observed in: (1) ICG: incorporated cell group or (2) ISC: incorporated single cells isolated within the chicken neural tube, or (3) CR: contiguous rosette with chick neural tube; (**C–G**) IF to detect key marker proteins in control (note, these non-grafted /sham operated embryos

*Figure 4 continued on next page*

*Figure 4 continued*

regenerate the dorsal neural tube by 2 days post-operation) and grafted embryos after 2 days incubation (**C**) ZO1, (**D**) SOX2, (**E**) PAX7, (**F**) P27, (**G**) ISLET1 (arrowheads indicate examples of candidate protein expressing cells in the chicken host tissue; arrow indicates examples of P27 and GFP co-expressing cells, dashed white boxes indicate area of magnification in next adjacent image), three transverse sections from each of at least three grafted chicken embryos were analysed for each marker combination and the proportions of SOX2/GFP+ ve, PAX7/GFP+ ve and P27/GFP+ ve cells in configurations 1, 2 or 3 were quantified, no ISLET1+ ve cells were found (this included three sections from three different grafts/embryos in which GFP+ ve cells were located within the chicken host Islet1 domain), each data point represents data from one histological section. Note that the small number of single cells (group 2) underlies lack of significant difference between groups (see Materials and Methods and metadata), data analysed with Mann–Whitney U test. Errors bars are ± SD. Scale bar = 50 μm.

The online version of this article includes the following source data and figure supplement(s) for figure 4:

**Source data 1.** Quantifications indicating proportions of SOX2/GFP + ve, PAX7/GFP + ve, P27/GFP + ve and ISLET1/GFP + ve cells in configurations (1) ICG: incorporated cell group or (2) ISC: incorporated single cells isolated within the chicken neural tube, or (3) CR: contiguous rosette with chick neural tube, 2 days after isochronic transplantation.

**Figure supplement 1.** Expression patterns of key marker proteins in human iPSC derived dorsal spinal cord rosettes cultured in vitro in parallel with those iso-chronically grafted into chick spinal cord.

**Figure supplement 1—source data 1.** Quantifications indicating proportions of cells expressing SOX2, PAX7, SNAI2, P27, ISLET1, and TFAP2α within a rosette differentiated from ChiPSC4-pmGFP from D6 cultured in parallel of transplantations experiments.

**Figure supplement 1—source data 2.** Quantifications indicating proportions of cells expressing SOX2, PAX7, SNAI2, P27, ISLET1, and TFAP2α within a rosette differentiated from ChiPSC4-pmGFP from D8 cultured in parallel of transplantations experiments.

**Figure supplement 1—source data 3.** Quantifications indicating proportions of cells expressing SOX2, PAX7, SNAI2, P27, ISLET1, and TFAP2α within a rosette differentiated from ChiPSC4-pmGFP from D11 cultured in parallel of transplantations experiments.

**Figure supplement 2.** Dorsal region of the chicken neural tube removal prior to grafting.

**Figure supplement 3.** Iso-chronic, iso-topic transplantation of dorsal neural tube from chicken embryos expressing a plasma membrane tagged with GFP into wild-type chicken embryos.

**Figure supplement 4.** Small groups of D6 human neural progenitors exhibit similar differentiation timing to whole neural rosettes following iso-chronic transplantation.

**Figure supplement 4—source data 1.** Quantifications indicating proportions of PAX7/GFP + ve, P27/GFP + veand ISLET1/GFP + ve cells in configurations (1) ICG or (2) ISC isolated within the chicken neural tube, 2 days after isochronic transplantation of smaller group of human neural progenitors.

**Figure supplement 4—source data 2.** Quantifications indicating proportions of PAX7/GFP + ve, P27/GFP + ve, and ISLET1/GFP + ve cells in configurations (1) ICG or (2) ISC: Incorporated Single Cells isolated within the chicken neural tube, 5 days after isochronic transplantation of smaller group of human neural progenitors.

*3*). Similar observations were also made in embryos in which the hemi-dorsal neural tube was removed but no rosette was transplanted prior to incubation and these embryos were processed as controls alongside grafted embryos (*Figure 4C–G*). This experimental design allowed comparison of differentiation between chicken dorsal neural tube and human D6 rosettes composed of similar populations of dorsal neural progenitors (*Figure 4—figure supplement 2*). Moreover, the D6 rosettes, which have brachial/thoracic character, were grafted into the corresponding region in the developing chicken embryo (opposite somites 10–16) providing an appropriate rostro-caudal regional context for isochronic evaluation of differentiation progression.

After 2 days incubation, grafted cells retained apico-basal polarity as indicated by localised ZO-1 and N-cadherin expression (*Aaku-Saraste et al., 1996*; *Hatta and Takeichi, 1986*; *Dady et al., 2012*; *Figure 4C*) and the majority of cells had formed rosette-like structures, which were adjacent but contiguous with the host neuroepithelium, while subsets of cells were incorporated within the host neuroepithelium, including some single cells (*Figure 4B, B'*). At this time, most grafted cells still expressed neural progenitor proteins (SOX2 and PAX7) (*Figure 4D, E*, *Figure 4—source data 1*) and the first P27 expressing neurons appeared (*Figure 4F*, *Figure 4—source data 1* and *Figure 4—figure supplement 1*, *Figure 4—figure supplement 1—source data 2*), but ISLET-1 expressing (dI3) interneurons were not yet detected (*Figure 4G*, *Figure 4—source data 1*). This is in good alignment with neural differentiation progression observed in the now D8 rosettes cultured in parallel in vitro (*Figure 4—figure supplement 1*, *Figure 4—figure supplement 1—source data 2*). Importantly, at this stage in the chicken embryo ISLET-1/dI3 expressing neurons are detected (*Figure 4G*, *Figure 4—figure supplement 3*; *Andrews et al., 2017*), consistent with the faster differentiation of the host

tissue and indicating that the pace of neural differentiation in grafted human rosettes is unaltered in this more rapidly differentiating environment.

One possibility to consider is that a community effect (*Gurdon, 1988*) operates within grafted rosettes, which overcomes differentiation signals provided by host tissues. We therefore analysed separately human rosette cells which were within a rosette formation abutting/continuous with the chicken neural tube (contiguous rosette, CR), a cell group directly incorporated into the chick neuroepithelium (incorporated cell group, ICG), or single cells incorporated within the chicken neuroepithelium (incorporated single cells, ISC) to determine if these different relationships with the chicken host cells influenced differentiation speed (*Figure 4B'*). However, no differences in differentiation were found between these three cell configurations (*Figure 4D–G*, *Figure 4—source data 1*). To address the further possibility that small groups or individual human cells may have moved from a rosette into the chicken neuroepithelium at a late time point and so only briefly experienced fuller integration into this host neuroepithelium, these experiments were repeated by initial grafting of small cell groups from dissociated rosettes (*Figure 4—figure supplement 4A and G*, *Figure 4—figure supplement 4—source data 1*). This approach further demonstrated that human cells in small groups and single human cells integrated into the chicken neuroepithelium do not differentiate more rapidly than in

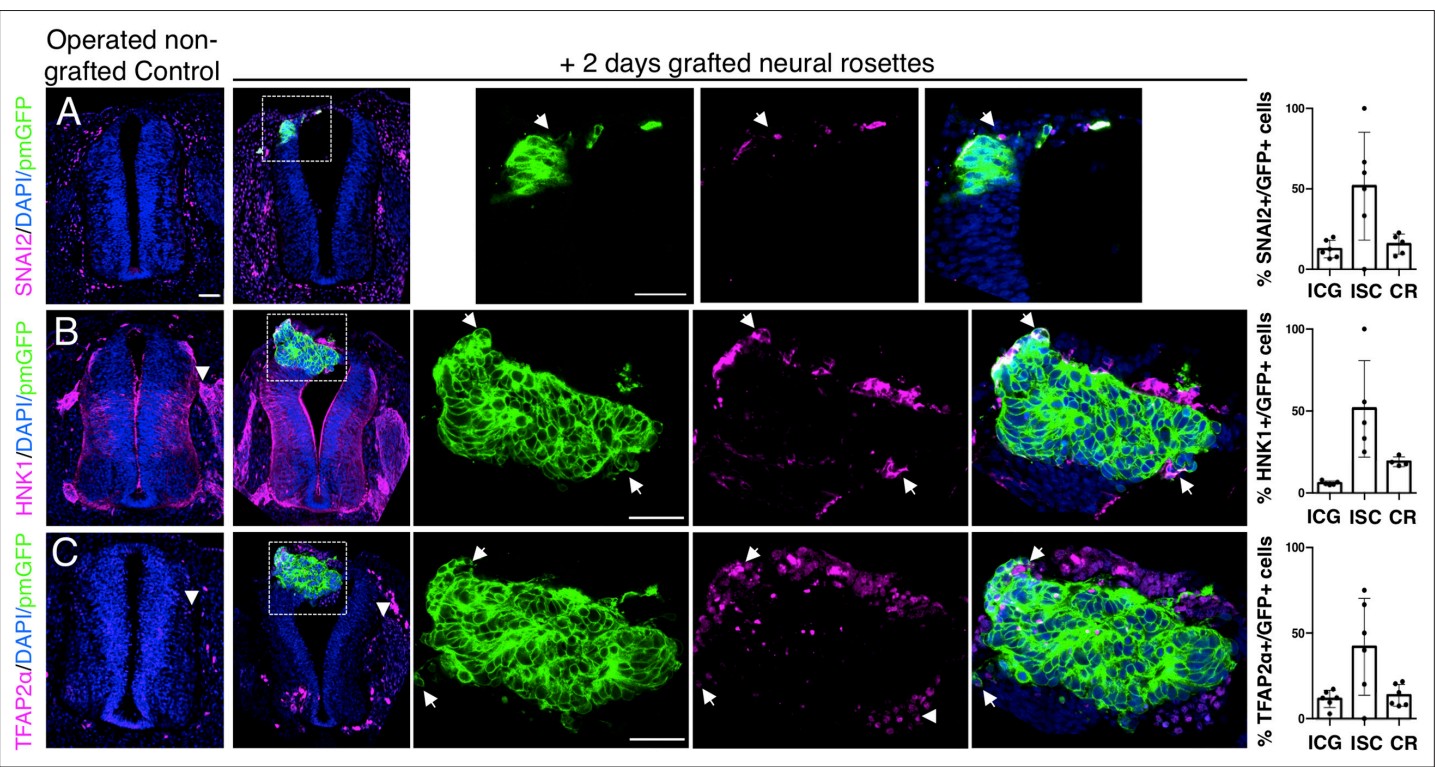

**Figure 5.** Iso-chronically transplanted hiPSC-derived neural rosettes exhibit cell intrinsic neural crest differentiation in the chicken embryonic environment. (**A–C**) IF to detect key neural crest marker proteins in control (non-grafted) and grafted embryos after 2 days incubation (**A**) SNAIL2, (**B**) HNK1, (**C**) TFAP2α (arrowheads indicate examples of candidate protein expressing cells in the chicken host tissue; arrows indicate examples of candidate protein and GFP co-expressing cells, note most cells at graft periphery, dashed white boxes indicate area of magnification in next adjacent image). Three transverse sections from each of at least three grafted chicken embryos were analysed for each marker and the proportions of SNAIL2/GFP+ ve, HNK1/GFP+ ve, and TFAP2α /GFP+ ve cells in cell configurations (1) ICG or (2) ISC within the chicken neural tube, or (3) CR with chick neural tube, each data point represents data from one histological section, note the small number of isolated cells (group 2) underlies lack of significant difference between groups (see section 'Materials and Methods'), data analysed with Mann–Whitney U test. Errors bars are ± SD. Scale bar = 50µm.

The online version of this article includes the following source data and figure supplement(s) for figure 5:

**Source data 1.** Quantifications indicating proportions of SNAI2/GFP+ ve, HNK1/GFP+ ve,and TFAP2α /GFP+ ve cells in configurations (1) ICG or (2) ISC isolated within the chicken neural tube, or (3) CR with chick neural tube, 2 days after isochronic transplantation.

**Figure supplement 1.** Transplanted human dorsal spinal cord rosettes express SOX10, a specific marker of differentiated neural crest 2 days after grafting.

in vitro conditions. These findings suggest that a community effect does not underpin the timing of human cell differentiation following transplantation into the chicken neural tube.

Analysis of neural crest differentiation in transplanted rosettes after 2 days further confirmed the asynchrony between human and host chicken cell differentiation. SNAI2-expressing neural crest cells were still found within these rosettes, while delamination of host neural crest was now complete, as indicated by absence of Snail2 positive cells in the chicken dorsal neural tube (*Figure 5A*, *Figure 5—source data 1*). Moreover, expression of migratory neural crest markers HNK1 and TFAP2α in the rosette periphery further aligned human rosette differentiation in vivo with that observed in vitro (*Figure 5B and C*, *Figure 5—source data 1* and *Figure 4—figure supplement 1*, *Figure 4—figure supplement 1—source data 2*): single integrated human cells were more likely to express markers of migrating neural crest cells HNK1 (*Figure 5B*, *Figure 5—source data 1*) and TFAP2α (*Figure 5C*, *Figure 5—source data 1*) exhibiting neural crest differentiation progression similar to that found in in vitro conditions (*Figure 4—figure supplement 1*, *Figure 4—figure supplement 1—source data 2*).

Intriguingly, despite this apparent lack of influence of the chick host on grafted human cells, we found some evidence that human cells altered chick neural crest differentiation. At this stage, chicken neural crest cells have normally delaminated and migrated to the dorsal root ganglia (DRGs) (*Le Douarin and Kalcheim, 1999*). However, close to integrated human cells chicken neural crest expressed early migration phase markers HNK1 (*Figure 5B*), TFAP2α (*Figure 5C*) and Sox10 (*Figure 5—figure supplement 1*) suggesting that the presence of human cells delays progression of the host neural crest differentiation program.

Together these findings indicate that after 2 days, differentiation in grafted human spinal cord rosettes progresses at the same pace as in vitro despite exposure to the more rapidly differentiating environment of the chicken host embryo.

## Longer term analysis reveals impaired differentiation of iso-chronically transplanted human neural progenitors

To address whether the differentiation pace in grafted human rosettes continues in synchrony with that in vitro, further analysis was undertaken 5 days after transplantation. At this time most human cells were incorporated within the host neuroepithelium, either as cell groups or single cells (*Figure 6A, B*) and the majority of cells remained dorsal neural progenitors expressing SOX2 and PAX7 (*Figure 6C, D*, *Figure 6—source data 1*). Importantly, neuronal differentiation still continued (*Figure 6E–I*, *Figure 6—source data 1*) as indicated by the presence of P27 positive human cells. However, the proportion of cells expressing P27 in transplanted rosettes decreased between days 2 and 5 and was also significantly reduced compared with their in vitro counterparts, where the proportion of P27 cells increased in this time frame (*Figure 6I*, *Figure 6—source data 1* and *Figure 4—figure supplement 1—source data 3*). The neuronal marker ISLET1 was first detected 5 days after grafting, both in vivo and in vitro (equivalent to D11 rosettes) (*Figure 6I*, *Figure 6—source data 1* and *Figure 4—figure supplement 1*, *Figure 4—figure supplement 1—source data 3*). However, the proportion of ISLET1 positive cells was significantly less in transplanted rosettes compared with their in vitro counterparts (*Figure 6I*, *Figure 6—source data 1* and *Figure 4—figure supplement 1—source data 3*). Moreover, the neural progenitor cell population continued to expand in vivo to a greater extent than in vitro, as indicated by the higher proportion of SOX2 and PAX7 expressing cells in transplanted rosettes (*Figure 6I*, *Figure 6—source data 1* and *Figure 4—figure supplement 1*, *Figure 4—figure supplement 1—source data 3*). The expansion of human neural progenitors was further indicated by the high proportion of cells expressing the proliferative marker Ki67 (*Figure 6G*, *Figure 6—source data 2*) while few apoptotic (cleaved caspase 3 expressing) cells were observed (*Figure 6H*, *Figure 6—source data 2*). Although it is formally possible that neuronal death takes place discretely between day 2 and day 5 and so was not detected in this analysis, together these findings indicate that neurogenesis attenuates in these initially iso-chronically transplanted rosettes while the neural progenitor population expands.

Furthermore, day 5 transplanted rosettes lacked the neural crest markers (SNAI2, HNK1 ,and TFAP2α), but these were detected in rosettes cultured in parallel in vitro (*Figure 7A-C*, *Figure 7—source data 1* and *Figure 4—figure supplement 1*, *Figure 4—figure supplement 1—source data 3*). This indicates that this later chicken embryo environment is no longer conducive to human neural crest differentiation. However, we note that human neural crest cells were still generated in one chimera

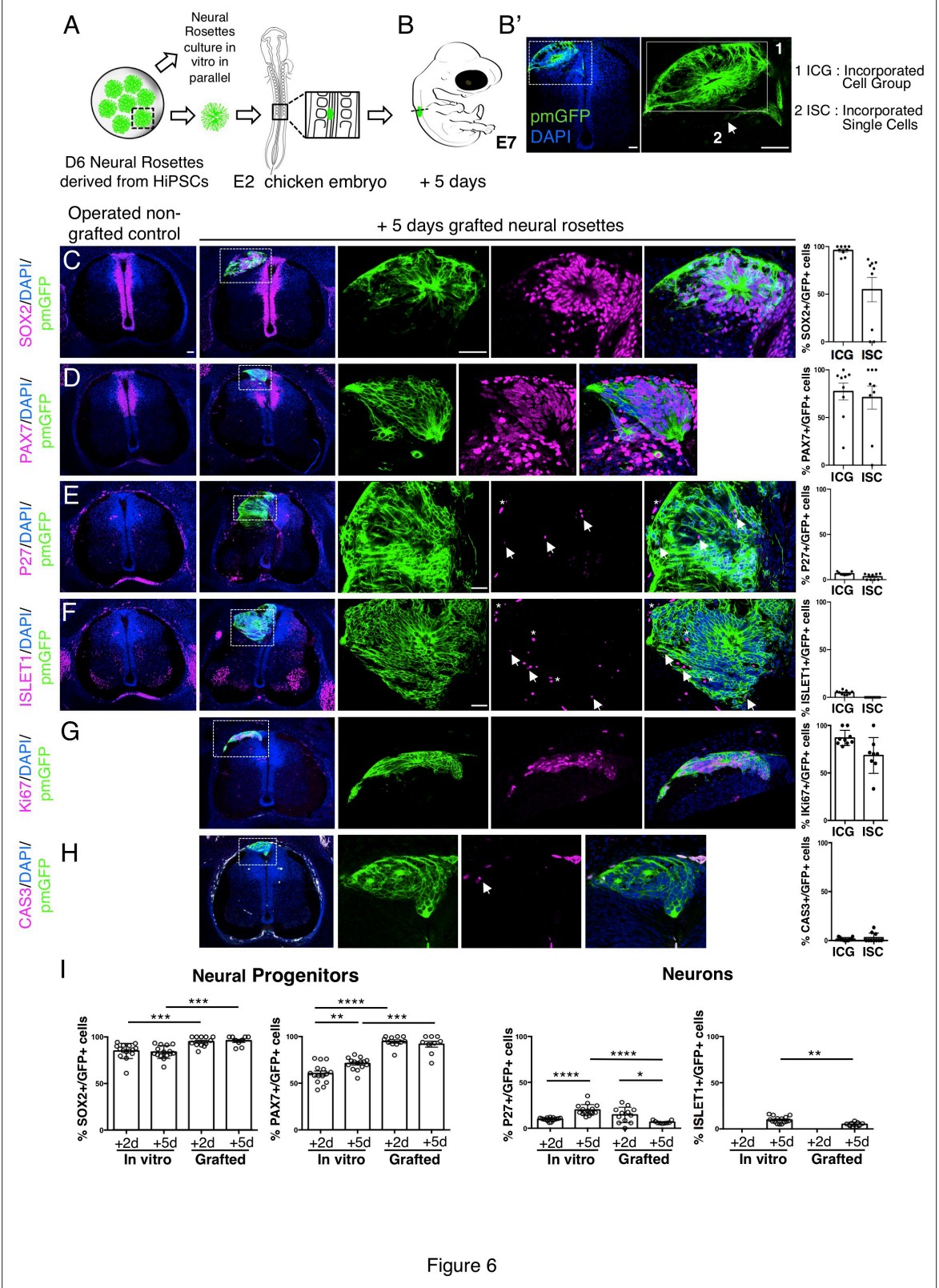

Figure 6

**Figure 6.** Long-term analysis reveals impaired neuronal differentiation and cell type specification in iso-chronically grafted human neural rosettes. (A) Schematic of human dorsal spinal cord neural rosette transplantation protocol and (B-B') transplanted cell configurations 5 days after transplantation; (C–H) IF to detect key marker proteins in control (non-grafted) and grafted chicken embryos after 5 days incubation, neural progenitor markers (C) SOX2, and (D) PAX7, post-mitotic neuronal (E) P27 and dorsal interneuron type 3, dl3s, (F) ISLET1 markers, proliferative cell marker (G) Ki67 and cell death

*Figure 6 continued on next page*

*Figure 6 continued*

marker (**H**) CASPASE-3 (arrows indicate examples of P27 or ISLET1 and GFP co-expressing cells, dashed white boxes indicate area of magnification in next adjacent image, asterisks indicate examples of host blood cells). Three transverse sections from each of at least three grafted chicken embryos were analysed for each marker and the proportions of SOX2/GFP+ ve, PAX7/GFP+ ve, P27/GFP+ ve, ISLET1/GFP+ ve, Ki67/GFP+ ve, and CAS-3/GFP + ve cells in configurations (1) ICG or (2) ISC within the chicken neural tube were quantified, each data point represents one section analysed, no significant differences were found between expression in ICG and ISC; (**G**) comparison of proportion of ChiPS4-pmGFP derived neural progenitors and neurons expressing SOX2, PAX7, P27, and ISLET1 in rosettes cultured in vitro in parallel with those transplanted into chicken embryos, assessed after 2 and 5 days, each data point represents a single rosette (five rosettes sampled from each to three independent differentiations), see *Figure 4—figure supplement 1* and proportions of these same marker proteins in transplanted rosettes after 2 and 5 days in chicken embryonic environment, each data point represents data from one section, with three sections from each of three grafted chick embryos, analysis Mann–Whitney U test. Errors bars are ± SD. p-values (see section 'Materials and Methods'). Scale bar = 50 μm.

The online version of this article includes the following source data for figure 6:

**Source data 1.** Quantifications indicating proportions of SOX2/GFP+ ve, PAX7/GFP+ ve, P27/GFP+ ve, and ISLET1/GFP + ve cells in configurations (1) ICG or (2) ISC isolated within the chicken neural tube, 5 days after isochronic transplantation.

**Source data 2.** Quantifications indicating proportions of Ki67/GFP+ ve and CAS3/GFP + ve cells in configurations (1) ICG: Incorporated Cell Group or (2) ISC: Incorporated Single Cells isolated within the chicken neural tube, 5 days after isochronic transplantation.

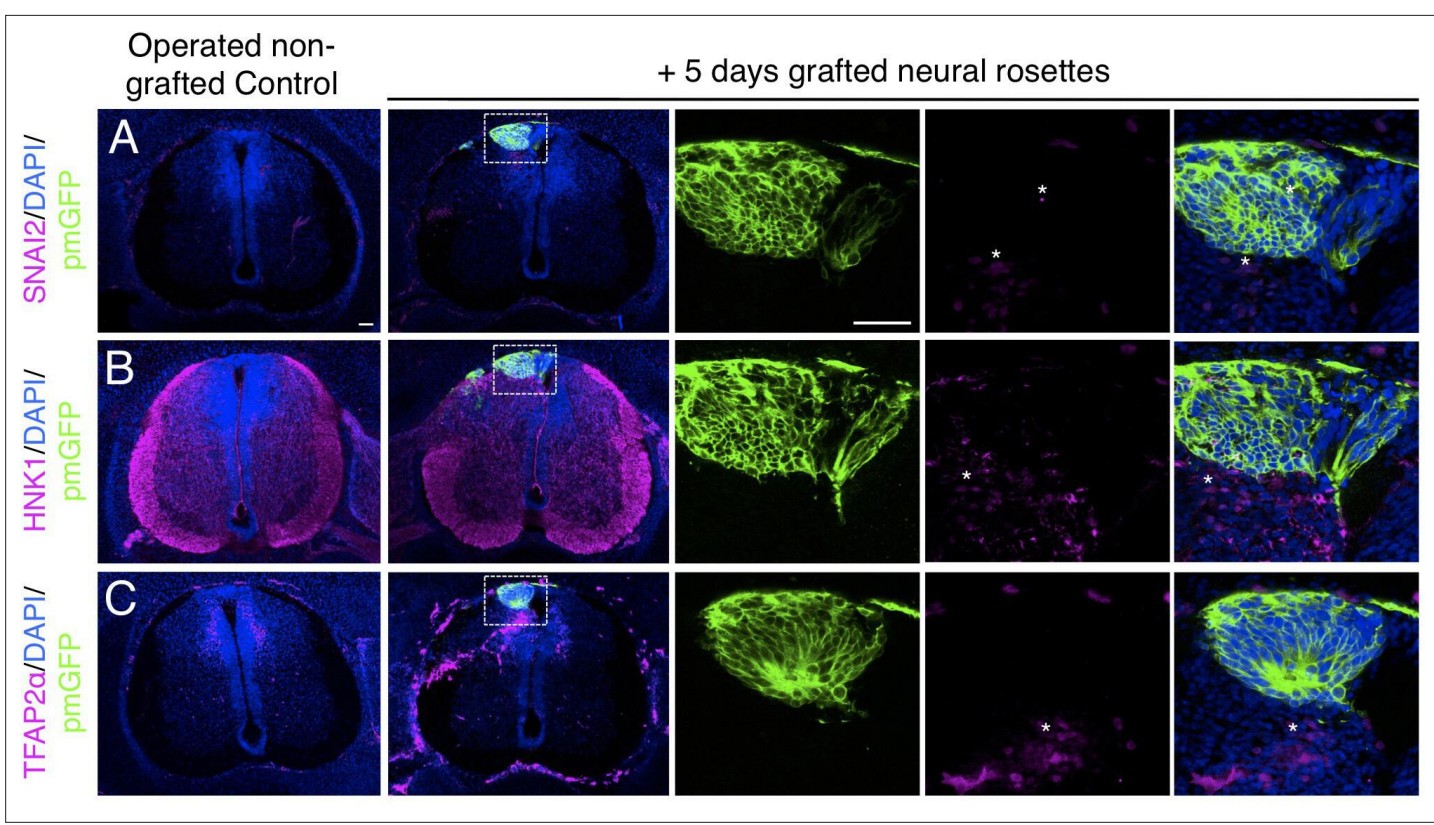

**Figure 7.** Long-term analysis reveals no neural crest differentiation in iso-chronically grafted human neural rosettes. (**A–C**) IF to detect key neural crest marker proteins in control (non-grafted) and grafted chicken embryos after 5 days incubation reveals no expression of neural crest progenitor markers in any cell configurations (**A**) SNAI2 (1: 0/860 GFP+ cells and 2: 0/70 GFP+ cells), (**B**) differentiated neural crest cells HNK1 (1: 0/956 GFP+ cells and 2: 0/57 GFP+ cells) and (**C**) TFAP2α (1: 0/1175 GFP+ cells and 2: 0/66 GFP+ cells) (asterisks indicate host blood cells). At least three chicken embryos and three cross-sections were analysed for each marker. Scale bar = 50 μm.

The online version of this article includes the following source data and figure supplement(s) for figure 7:

**Source data 1.** Quantifications indicating proportions of SNAI2/GFP+ ve, HNK1/GFP+ ve, and TFAP2α /GFP+ ve cells within the chicken neural tube, 5 days after isochronic transplantation.

**Figure supplement 1.** Ectopically positioned human dorsal spinal cord rosette reveals retention of ability to differentiate into neural crest.

excluded due to open neural tube defects (*Figure 7—figure supplement 1*), suggesting that human neural rosettes retain the potential for such differentiation in an appropriate environment.

Overall, this longer-term analysis of differentiation in transplanted human rosettes revealed that rather than proceeding more rapidly in the chicken embryo, the apparently intrinsic differentiation pace of human dorsal neural progenitors and neural crest is not sustained and cells in the human rosettes largely remain in a neural progenitor cell state.

## Heterochronic transplantation of human neural progenitors into a developmentally older chicken neuroepithelium ultimately promotes human neurogenesis

To assess whether the differentiation pace of human neural rosette cells can be altered by transplantation into a more developmentally advanced chicken neuroepithelium, D6 neural rosettes were next transplanted into the dorsal spinal cord of older chicken embryos (E3, stage HH 20–21) (*Figures 8A and 9A*): a stage at which the chicken neuronal differentiation programme is well established (*Andrews et al., 2017*). After 2 days, most of the transplanted human cells retained expression of the dorsal progenitor marker PAX7 (*Figure 8C*, *Figure 8—source data 1*) and some also expressed the post-mitotic/neuronal marker P27 (*Figure 8D*, *Figure 8—source data 1*). Moreover, as found in vitro, ISLET1 was also not detected in these hetero-chronically transplanted human cells at this time (*Figure 8E*, *Figure 8—source data 1* and *Figure 4—figure supplement 1—source data 2*). Furthermore, while the dorsal glial marker NFIA was now expressed in the surrounding chicken neuroepithelium (and see *Deneen et al., 2006*), this is not detected in the transplanted human cells (*Figure 8F*, *Figure 8—source data 1*). These findings indicate that the host chicken environment does not impact human neural progenitor differentiation over this timeframe.

Strikingly, 5 days after hetero-chronic transplantation human neural rosettes now exhibited fewer PAX7-expressing cells and increased P27 and ISLET1 expressing cells in comparison to the same age rosettes in vitro (or following isochronic transplantation) (*Figure 9C–F*, *Figure 9—source data 1*, *Figure 4—source data 1*, *Figure 4—figure supplement 1—source data 3*). All GFP expressing cells were located within the spinal cord either as incorporated cell groups or as single cells. While the latter exhibited reduced PAX7 and increased P27 and ISLET1 these single cells were located close to the periphery of the rosette consistent with neuronal birth taking place in this position as cells emerge from the rosette. Overall, these data show that transplanted human cells can respond to signals provided by this more differentiated chicken host environment and indicate that the onset of human neurogenesis can be advanced if neural progenitors are exposed to an older differentiating environment over a period longer than 2 days.

## Discussion

This study presents a novel in vitro human neural rosette assay that recapitulates the temporal sequence of dorsal spinal cord differentiation observed in vertebrate embryos. The more rapid progression of this differentiation programme in vitro than in the human embryo suggests that it represents the intrinsic differentiation pace of this cell population, while lacking signalling dynamics that influence the normal duration of steps in spinal cord differentiation. The initial retention of in vitro differentiation pace in human rosettes transplanted into the faster differentiating chicken embryonic spinal cord, either iso-chronically or hetero-chronically, further supports the notion that this reflects species-specific cell intrinsic timing. Importantly, long-term analysis of transplanted rosettes revealed that the iso-chronic environment was insufficient to sustain the human neural differentiation programme, while the more developmentally advanced hetero-chronic environment accelerated neurogenesis. These findings show that after a delay, which likely reflects previously reported slower protein turnover and cell cycle of human neural progenitors, differentiation of human cells can be influenced by the chicken embryonic environment. Moreover, hetero-chronic grafting into a more differentiated neural tube while increasing neuron production in comparison with neural rosettes in vitro, did not alter the programme of human neural differentiation, which progressed to a neurogenic phase, while the host embryo advanced into gliogenesis (*Figure 10*). These experiments provide evidence for cell intrinsic constraint on human differentiation pace and highlight the importance of timely extrinsic signalling for progression through an intrinsic human neural differentiation programme.

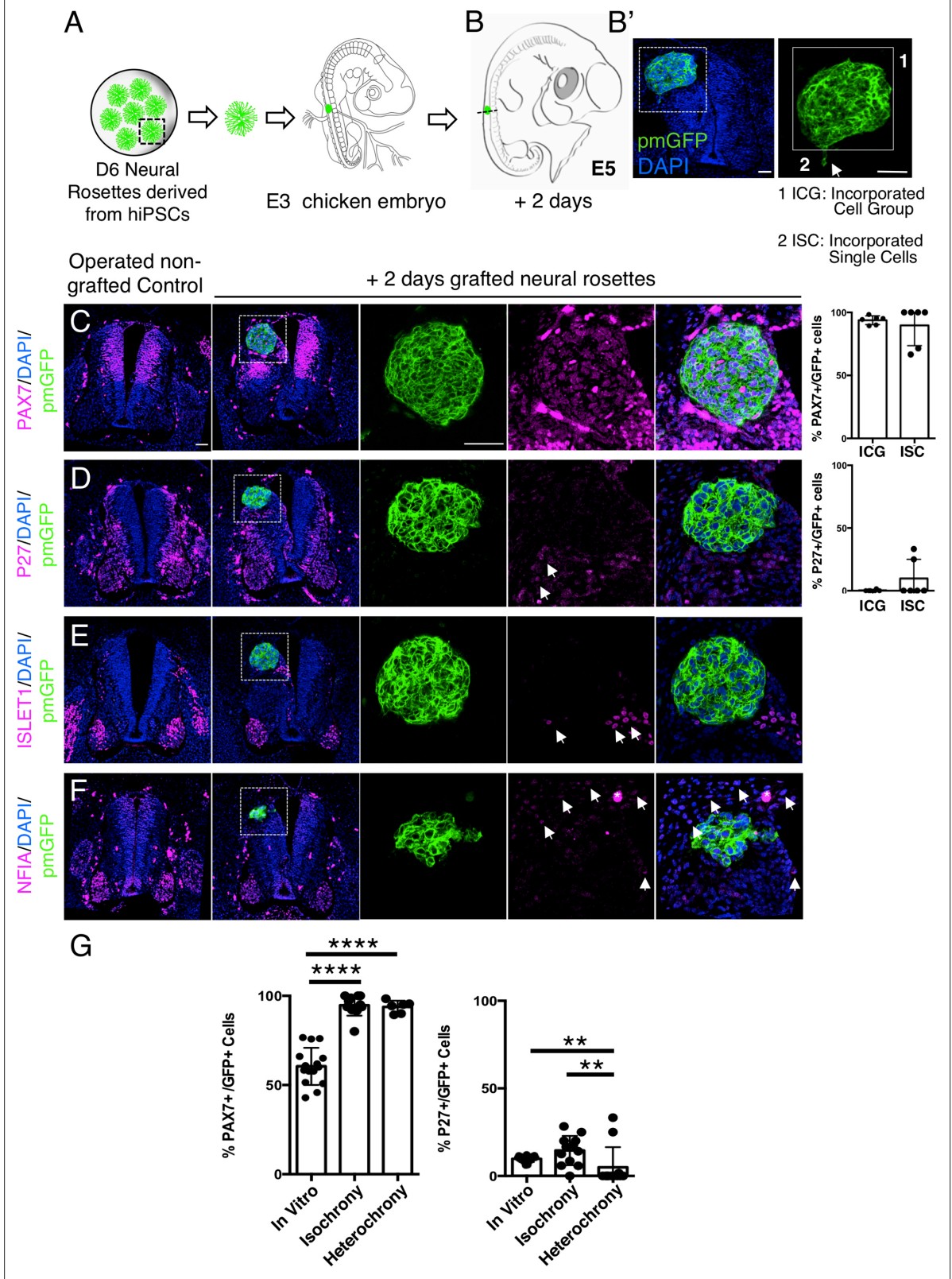

**Figure 8.** Hetero-chronically transplanted hiPSC-derived neural rosettes exhibit reduced neuronal differentiation despite exposure to more rapidly developing chicken embryonic environment. (**A**) Schematic of human dorsal spinal cord neural rosette transplantation experiment; (**B-B′**) Two days after grafting human cells were observed in: (1) ICG or (2) ISC within the chicken neural tube. Note the E3 chicken embryo has a less regulative response to hemi-dorsal tube removal, this did not alter expression of differentiation markers, but distorted tissue morphology in some operated non-grafted

*Figure 8 continued on next page*

*Figure 8 continued*

controls; (**C–F**) IF to detect key marker proteins in control (note, these non-grafted /sham operated embryos regenerate the dorsal neural tube by 2 days post-operation) and grafted embryos after 2 days incubation (**C**) PAX7, (**D**) P27, (**E**) ISLET1, and (**F**) glial marker, NFIA (dashed white boxes indicate area of magnification in next adjacent image), three transverse sections from each of at least three grafted chicken embryos were analysed for each marker combination and the proportions of PAX7/GFP + ve and P27/GFP + ve cells in configurations 1 and 2 were quantified, each data point represents data from one histological section(see 'Materials and Methods'); (**E**) comparison of proportion of ChiPS4-pmGFP derived neural progenitors and neurons expressing PAX7 and P27 in rosettes cultured in vitro in parallel with those transplanted iso-chronically and hetero-chronically into chicken embryos, assessed after 2 days, each data point represents a single rosette (five rosettes sampled from each to three independent differentiations), see *Figure 4—figure supplement 1*, and proportions of these same marker proteins in transplanted rosettes after 2 days in chicken embryonic environment, each data point represents all data from one section, with three sections from each of three grafted chicken embryos, analysis Mann–Whitney U test. Errors bars are± SD. p-values (see 'Materials and Methods'). Scale bar = 50 µm.

The online version of this article includes the following source data for figure 8:

**Source data 1.** Quantifications indicating proportions of PAX7/GFP+ ve, P27/GFP+ ve, ISLET1/ GFP + ve, and NFIA/GFP+ ve cells in configurations (1) ICG or (2) ISC isolated within the chicken neural tube, 2 days after heterochronic transplantation.

In vitro differentiation of human pluripotent cell derived NMPs in minimal medium containing low level RA (*Verrier et al., 2018*) was sufficient to generate self-organizing dorsal spinal cord rosettes composed of neural crest and neural progenitors. We demonstrated that such structures reproducibly differentiate in a specific temporal sequence which follows that observed in the embryonic dorsal spinal cord (*Figure 3I*), with neural crest differentiating first, then dorsal-interneurons (dI1/3/5 s) and finally glial cells (*Duband et al., 2015*; *Jiang et al., 2009*; *Andrews et al., 2017*; *Deneen et al., 2006*). One peculiarity in this assay, the lack of dI2/4 s interneurons, may be explained by a requirement for more nuanced exposure to BMPs (*Andrews et al., 2017*; *Gupta et al., 2018*; *Duval et al., 2019*), which, although endogenously expressed in this assay, are not exogenously provided in this protocol (*Verrier et al., 2018*).

Several lines of evidence indicate that the in vitro human dorsal spinal cord rosettes exhibit an intrinsic differentiation pace. First, the timing of neural progenitor differentiation was found to be more rapid in vitro than in the human embryo. This was indicated by the earlier appearance of ISLET1+/HB9- expressing neurons and of NFIA expressing gliogenic progenitors in neural rosettes than in the human embryonic dorsal spinal cord; indeed, the period from migratory neural crest emergence to gliogenesis onset was ~1.5 weeks earlier in vitro than in the human embryo (*Figure 3I*). Although it is possible that variation between human embryos contributes to this discrepancy, this may reflect longer time spent in expansive neural progenitor phases in the embryo, where this leads to generation of a much larger structure than a rosette. This raises the interesting possibility that overall differentiation pace, beyond that governed by cell intrinsic factors, might be influenced by and scale with the size of a self-organising cell group.

Secondly, iso-chronic or hetero-chronic transplantation of human neural rosettes into the chicken neural tube did not, in the first 2 days, advance the timing of human neural differentiation (even in individual human cells incorporated within the chicken neuroepithelium). Moreover, by 5 days iso-chronic conditions proved insufficient to sustain human neural differentiation, while more developmentally advanced hetero-chronic conditions now promoted human neural differentiation. As a slow response takes place regardless of the extrinsic cues provided these findings suggest that the ability of transplanted human cells to respond to the chicken environment is constrained by species-specific cell intrinsic properties. Such constraint is consistent with the finding that human ventral neural progenitors have a longer cell cycle and differentiate more slowly than equivalent mouse cells cultured in similar conditions in vitro (*Rayon et al., 2020*). This has been shown to reflect species-specific differences in general protein stability (*Rayon et al., 2020*), may be indicative of slower human biochemical reactions (*Matsuda et al., 2020*) and metabolism (*Diaz-Cuadros et al., 2021*), and explain the delayed response of human cells to signals provided by the chicken environment. Indeed, transcriptional onset of some differentiation genes may require exposure to appropriate signals in a specific cell cycle phase (*Pauklin and Vallier, 2013*; *Pauklin et al., 2016*) and involve defined changes in replication timing at the early G1 timing decision point (*Dimitrova and Gilbert, 1999*; *Hiratani et al., 2008*). Perhaps related to this is the finding that while human and mouse neural progenitor cells similarly transduce exogenously provided signalling molecule sonic hedgehog, human cells are slower to transcribe some key downstream ventral patterning genes *OLIG2* and *NKX6.1* (*Rayon et al., 2020*).

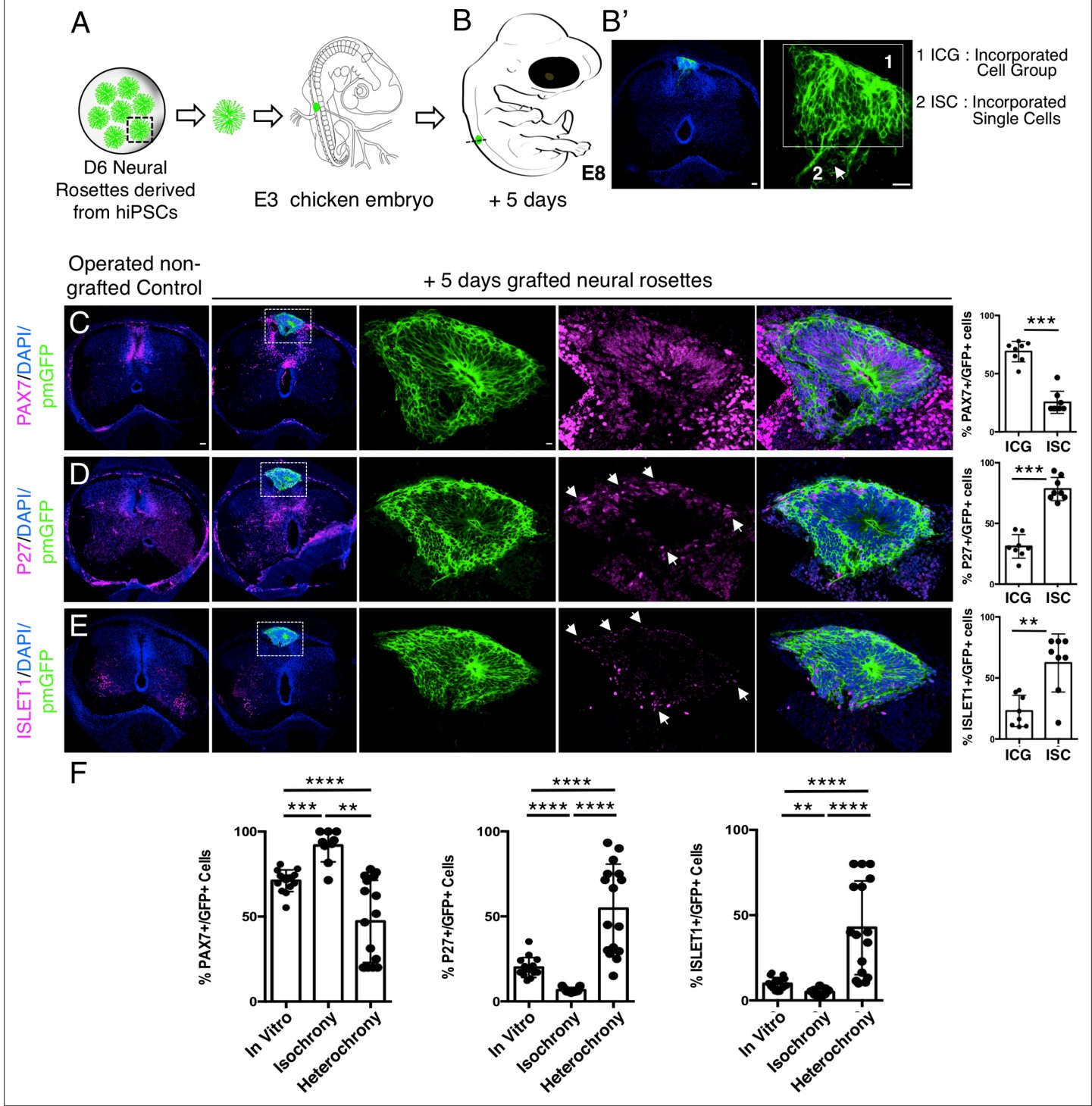

**Figure 9.** Longer-term analysis reveals increased neuronal differentiation in hetero-chronically transplanted human neural rosette. (**A**) Schematic of human dorsal spinal cord neural rosette transplantation experiment; (**B-B'**) Five days after grafting human cells were observed in: (1) ICG or (2) ISC within the chicken neural tube. (**C–E**) IF to detect key marker proteins in control (non-grafted) and grafted chicken embryos after 5 days incubation, (**C**) neural progenitor marker PAX7, (**D**) post-mitotic neuronal P27, and (**E**) dorsal interneuron type 3, dI3s, marker ISLET1, (arrows indicate examples of P27 or ISLET1 and GFP co-expressing cells, dashed white boxes indicate area of magnification in next adjacent image). Three transverse sections from each of at least three grafted chicken embryos were analysed for each marker and the proportions of PAX7/GFP+ ve, P27/GFP+ ve, and ISLET1/GFP+ ve cells in configurations one or two were quantified, each data point represents counts from one histological section; (**F**) comparison of proportion of ChiPS4-pmGFP derived neural progenitors and neurons expressing PAX7, P27, or ISLET1 in rosettes cultured in vitro in parallel with those transplanted iso-chronically and hetero-chronically into chicken embryos, assessed after 5 days, each data point represents data from a single rosette (five rosettes

*Figure 9 continued on next page*

*Figure 9 continued*

sampled from each of three independent differentiations), see *Figure 4—figure supplement 1*, and proportions of these same marker proteins in transplanted rosettes after 5 days in the chicken embryonic environment, each data point represents all data from one section, with three sections from each of three grafted chick embryos, analysis Mann–Whitney U test. Errors bars are ± SD. p-values (see section 'Materials and methods'). Scale bar = 50 μm.

The online version of this article includes the following source data for figure 9:

**Source data 1.** Quantifications indicating proportions of PAX7/GFP+ ve, P27/GFP+ ve, and ISLET1/ GFP + ve cells in configurations (1) ICG or (2) ISC isolated within the chicken neural tube, 5 days after heterochronic transplantation.

This may indicate that initiation of transcription of such genes requires transit through the cell cycle and that this contributes to the slower human neural differentiation pace.

In vertebrate embryos, the timing of specific phases of neural differentiation is orchestrated by signals from adjacent tissues, which drive and coordinate axial differentiation. This includes: adjacent somites (*del Corral et al., 2002*, *Diez del Corral et al., 2003*), underlying notochord (*Placzek et al., 1990*), and overlying ectoderm/roof plate (*Liem et al., 1997*; *Lee et al., 2000*). Clearly, the localised and temporally controlled characteristics of such regulatory influences are lacking in in vitro assays. The importance of timely exposure to extrinsic cues is underscored by the distinct responses observed following iso-chronic and hetero-chronic transplantation into the chicken embryonic environment. The expression of neural progenitor proteins SOX2 and PAX7 in almost all iso-chronically transplanted human rosette cells after 5 days along with proliferation marker Ki67 and the lack of apoptosis, supports the notion that that these cells are not exposed to sufficiently high levels of differentiation factors. In contrast, in the hetero-chronic condition host neurogenesis is well established and the significant increase in neuronal differentiation in these rosettes by 5 days may therefore reflect longer exposure to higher levels of neuronal differentiation factors. While minimal RA is provided in vitro, the RA synthesising enzyme *Raldh2* is expressed within the chicken neural tube including by the roof plate as well as the meninges enclosing the spinal cord at the time of hetero-chronic grafting E3 (HH19/20) (*Berggren et al., 1999*).

It is notable that the increased neuron production in heterochronic conditions follows the sequence of the human neural differentiation programme even when the host chicken embryo has progressed further and has been gliogenic for days 3–5 (from E5 to E8) (*Deneen et al., 2006*, see *Figure 10*). This indicates that the host environment is sufficient to support human neural differentiation progression and that this does not involve obviating the neurogenic phase. This is consistent with a related study in which mouse Olig2-expressing progenitors follow neurogenic and subsequent gliogenic phases after grafting into the E2 chicken embryo, and older gliogenic progenitors could not be reprogrammed back to a neurogenic phase in this younger chicken host (*Mukouyama al., 2006*). Our data here support operation of a similar intrinsic neural differentiation programme in human cells. It remains unclear whether the increased neuron production in heterochronic conditions represents a change in timing of such an intrinsic differentiation programme or simply the earlier exit of cells from the cell cycle: whether precocious neurogenesis is followed by earlier gliogenesis onset in comparison to in vitro conditions would require analyses at a still later stage. However, support for the possibility of such an acceleration of human differentiation timing is provided by recent transcriptomic analysis of neural differentiation in in vitro co-cultures of human ESCs and mouse EpiSCs, which found acceleration of human neural gene expression in the presence of increasing numbers of differentiating mouse EpiSCs (*Brown et al., 2021*).

This interpretation places emphasis on the signalling environment and predicts that iso-chronically grafted rosettes, which at day 5 exhibit reduced neurogenesis and arrested neural crest differentiation in comparison with in vitro conditions, will eventually differentiate. However, appropriate cues within the chicken embryo, in particular, for the neural crest, may no longer be present. Previous studies have used xenografting assays to evaluate the behaviour of human cells generated in vitro (*Valensi-Kurtz et al., 2010*; *Frith et al., 2018*) or for potential long-term regenerative therapy purposes using either neural progenitor cells or differentiated neurons (e.g., *Kumamaru et al., 2018*; *Kumamaru et al., 2019*; *Linaro et al., 2019*). Our findings emphasise a requirement for appropriate extrinsic cues for differentiation progression, although we cannot rule out inhibitory signals also acting on grafted human neural cells. Most striking was the continued expansion of iso-chronically transplanted neuroepithelial cells; a behaviour reminiscent of neural cancers such as glioblastoma that share a

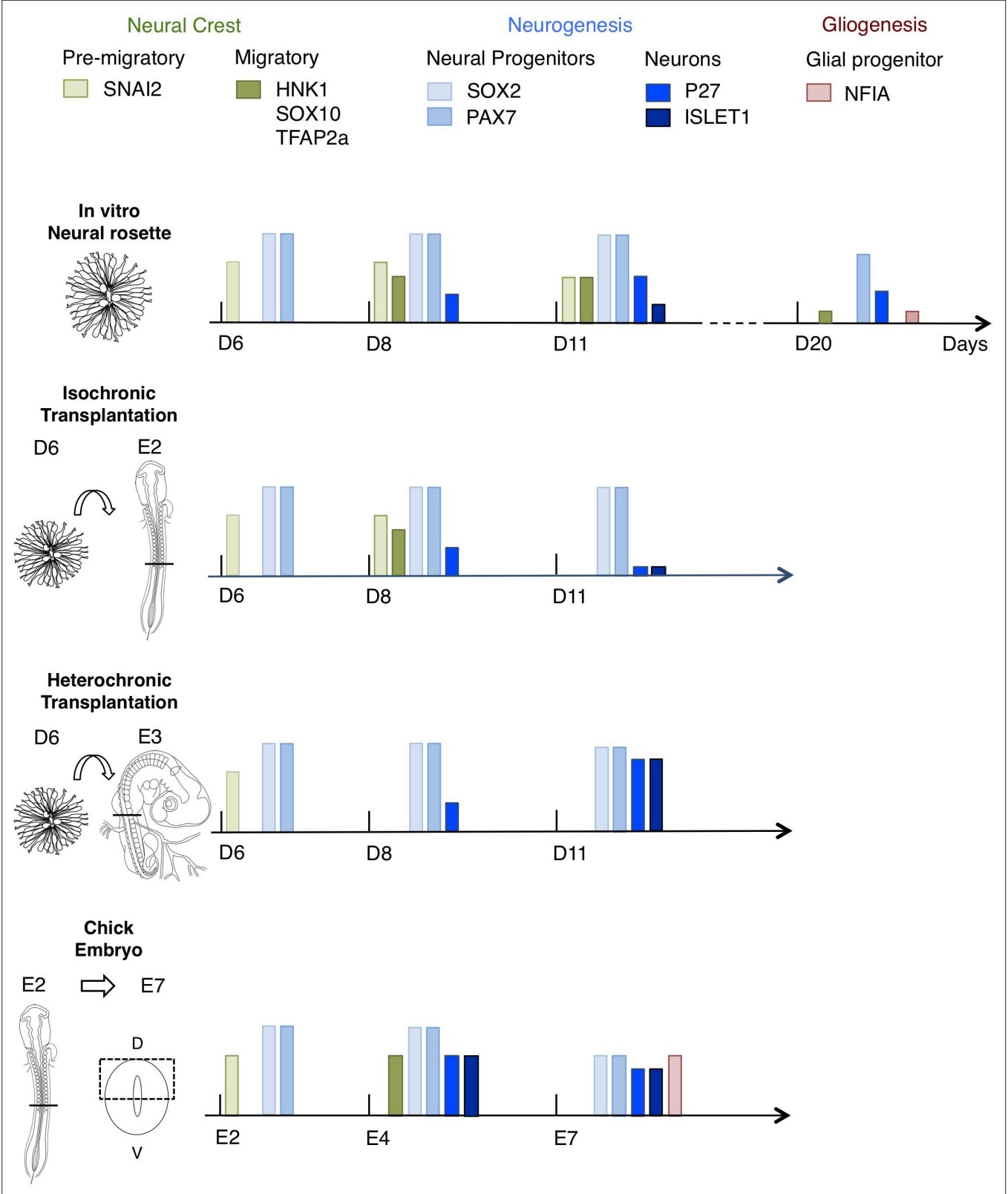

**Figure 10.** Schematic summary comparison of dorsal spinal cord differentiation in rosettes in vitro and following iso-chronic or hetero-chronic grafting and in the developing chicken embryo. The timing of differentiation from a dorsal spinal cord neural rosette composed of SOX2/ PAX7 neural progenitors and pre-migratory neural crest cells expressing SNAI2, into neurons (expressing P27 or ISLET1) and migratory neural crest cells (expressing HNK1, TFAP2a and SOX10) and subsequent gliogenesis in these distinct conditions is indicated (D is day of rosette differentiation from NMP-Like cells

*Figure 10 continued on next page*

*Figure 10 continued*

in vitro, days from transplantation of D6 rosettes is shown for transplantation experiments, and E is for Embryonic day of development in the chicken embryo). Note neural crest was not detected following hetero-chronic transplantation after 2 days (**D8**) and was not assessed at 5 days (**D11**). All empty slots represent no marker detection.

similar neural progenitor gene expression profile (*Agnihotri et al., 2013*; *Hassn Mesrati et al., 2020*). In addition to providing novel insight into the regulation of human neural differentiation, this assay may therefore also offer an opportunity to study cancer origins as well as inform generation of human organs in animal models and cell-based therapeutic approaches to neural injury and disease.

# Materials and methods

**Key resources table**

| Reagent type (species) or resource | Designation | Source or reference | Identifiers | Additional information |
|---|---|---|---|---|
| Genetic reagent (*Homo sapiens*) | Human embryo tissue (stage CS12, CS13, CS16 CS18) | Human Developmental Biology Resource (HDBR: https://www.hdbr.org) | | |
| Cell line (*Homo sapiens*) | ChiPSC4 line | Cellartis, AB and HPCF, University of Dundee | RRID:CVCL_RM97 | Cell line maintained in HPCF University of Dundee (Dr Lindsay Davidson) |
| Cell line (*Homo sapiens*) | hESCs line (H9) | WiCell | RRID:CVCL_9773 | |
| Cell line (*Homo sapiens*) | hESCs line (H1) expressing DCX$^{Cit/Y}$ | Allen Institute and WiCell | PMID:28094016 | Dr Boaz Levi |
| Biological sample (Chicken eggs) | myr-GFP Chicken embryos | National Avian Research Facility Roslin Institute, University of Edinburgh | PMID:25812521 | |
| Antibody | anti-SOX2 (Goat polyclonal) | Immune Systems | Cat# GT15098, RRID: AB_2732043 | IF (1:200) |
| Antibody | anti-ZO1 (Rabbit polyclonal) | Invitrogen ThermoFisher | Cat# 40–2200 RRID:AB_2533456 | IF (1:500) |
| Antibody | anti-N-Cadherin (Mouse monoclonal) | Sigma-Aldrich | Cat# C3865 RRID:AB_262097 | IF (1:500) |
| Antibody | anti-TUJ1 (Rabbit polyclonal) | Sigma-Aldrich | Cat# T2200 RRID:AB_262133 | IF (1:1000) |
| Antibody | anti-TUJ1 (Mouse monoclonal) | BioLegend | Cat# 801,201 RRID:AB_2313773 | IF (1:1000) |
| Antibody | anti-OLIG2 (Rabbit polyclonal) | Millipore | Cat# AB9610 RRID:AB_570666 | IF (1:200) |
| Antibody | anti-LHX2 (Rabbit polyclonal) | Sigma-Aldrich | Cat# HPA000838 RRID:AB_2666109 | IF (1:200) |
| Antibody | anti-ISLET1 (Rabbit polyclonal) | Abcam | Cat# Ab20670 RRID:AB_881306 | IF (1:200) |
| Antibody | anti-HB9 (Mouse monoclonal) | Developmental Studies Hybridoma Bank (DSHB) | Cat# 81.5C10 RRID:AB_2145209 | IF (1:10) |
| Antibody | anti-SNAI2 (Rabbit polyclonal) | Cell Signalling Technology | Cat# 9,585 RRID:AB_2239535 | IF (1:200) |
| Antibody | anti-SOX10 (Goat polyclonal) | R&D Systems | Cat# AF2864-SP RRID:AB_442208 | IF (1:200) |
| Antibody | anti-TFAP2α (Mouse monoclonal) | Santa Cruz | Cat# sc8975 RRID:AB_2240215 | IF (1:10) |

*Continued on next page*

*Continued*

| Reagent type (species) or resource | Designation | Source or reference | Identifiers | Additional information |
|---|---|---|---|---|
| Antibody | anti-HNK1 (Mouse monoclonal) | Developmental Studies Hybridoma Bank (DSHB) | Cat# 3H5 RRID:AB_2314644 | IF (1:10) |
| Antibody | anti-P27 (Mouse monoclonal) | BD Biosciences | Cat# BD610642 RRID:AB_397637 | IF (1:100) |
| Antibody | anti-activated CASPASE3 (Mouse monoclonal) | Sigma-Aldrich | Cat# MAB10753 RRID:AB_2904207 | IF (1:200) |
| Antibody | anti-Ki67 (Rat monoclonal) | Invitrogen ThermoFisher | Cat# 14-5698-82 RRID:AB_10854564 | IF (1:200) |
| Antibody | anti-NFIA (Mouse monoclonal) | Sigma-Aldrich | Cat# HPA008884 RRID:AB_1854421 | IF (1:200) |
| Antibody | anti-PAX7 (Mouse monoclonal) | Developmental Studies Hybridoma Bank (DSHB) | Cat# PAX7 RRID:AB_2299243 | IF (1:10) |
| Antibody | anti-PAX6 (Mouse monoclonal) | R&D Systems | Cat# MAB1260 RRID:AB_2159696 | IF (1:100) |
| Antibody | anti-goat 594 (Donkey clonality unknown) | ThermoFisher | Cat# A-11058 RRID:AB_142540 | IF (1:500) |
| Antibody | anti-mouse 488 (Donkey clonality unknown) | ThermoFisher | Cat# A-21202 RRID:AB_141607 | IF (1:500) |
| Antibody | anti-mouse 568 (Donkey clonality unknown) | ThermoFisher | Cat# A-10037 RRID:AB_2534013 | IF (1:500) |
| Antibody | anti-rabbit 488 (Donkey clonality unknown) | ThermoFisher | Cat# A-21206 RRID:AB_2535792 | IF (1:500) |
| Antibody | anti-rabbit 568 (Donkey clonality unknown) | ThermoFisher | Cat# A-10042 RRID:AB_2534017 | IF (1:500) |
| Recombinant DNA reagent | aPX1 (Plasmid) | Dr Timothy A. Sanders | | Dr Timothy A. Sanders, University of Chicago |
| Peptide, recombinant protein | bFGF | PreproTech | Cat# 100-18B | 20 ng/ml |
| Peptide, recombinant protein | Noggin | PreproTech | Cat# 120–10 C | 50 ng/ml |
| Commercial assay or kit | Neon Transfection System 10 µl Kit | ThermoFisher | Cat# MPK1025 | |
| Commercial assay or kit | pENTR /D-TOPO Cloning Kit | ThermoFisher | Cat# K240020 | |
| Commercial assay or kit | Gateway LR Clonase Enzyme mix | ThermoFisher | Cat# 11791019 | |
| Chemical compound, drug | All-trans-Retinoic Acid | Sigma-Aldrich | Cat# R2625 | 100 nM |
| Chemical compound, drug | CHIR 99021 | Tocris | Cat# 4,953 | 3 µM |
| Chemical compound, drug | SB431542 | Tocris | Cat# 1,614 | 10 µM |
| Chemical compound, drug | Y-27632 | Tocris | Cat# 1,254 | 10 µM |
| Software, algorithm | GraphPad Prism | GraphPad Software | RRID:SCR_002798 | https://www. graphpad. com/ |
| Other | DAPI stain | Invitrogen | D1306 | (1 µg/ml) |
| Other | Neurobasal Medium, minus phenol red | Gibco | Cat# 12348017 | 500 ml |
| Other | B-27 Supplement | Gibco | Cat# 17504044 | (1:50) |

Continued

| Reagent type (species) or resource | Designation | Source or reference | Identifiers | Additional information |
| --- | --- | --- | --- | --- |
| Other | N-2 Supplement | Gibco | Cat# A1370701 | (1:100) |
| Other | Glutamax | Gibco | Cat# 35050061 | (1:100) |
| Other | Geltrex | Life Technologies | Cat# A1413302 | |
| Other | TryPLE select | ThermoFisher | Cat# 12563011 | |

## Human pluripotent cell authentication and quality control

Human ES cell lines were obtained from WiCell, H9 (WA09) and from the Allen Institute/ WiCell (Dr Boaz Levi) (*Yao et al., 2017*), H1-DOUBLECORTIN (DCX)-CITRINE line (AI03e-DCX-YFP (aka DCX$^{Cit/Y}$ H1), and the human iPSC line ChiPS4 was provided by Cellartis AB (now Takara Bio). Suppliers (WiCell and Cellartis AB) provided documentation confirming cell identity. STR DNA profiling for ChiPS4, ChiPS4-pmGFP, H9 (WA09) and H1-DCX-CITRINE confirmed identity of these lines in our laboratory.

All cell lines were expanded and banked on arrival. Prior to experiments representative lots of each cell bank were thawed and tested for post-thaw viability and to ensure sterility and absence of mycoplasma contamination. Sterility testing was performed by direct inoculation of conditioned medium into tryptic soya broth and soya bean casein broth and no contamination was observed. Mycoplasma testing was carried out by DAPI staining of fixed cultures and using the mycoalert mycoplasma detection kit (Lonza). We confirm that the cell lines used in this study were free of mycoplasma and other microorganisms.

H9 (WA09) hES cells were supplied at passage 24, the cells were thawed, and cell banks prepared at passage 29: for experiments the cells were used between passage 29 and 39. H1-DOUBLECORTIN-CITRINE cells were supplied at passage 63 and cell banks prepared at passage 66: for experiments the cells were used between passage 66 and 76. ChiPS4 hiPS cells were supplied at passage nine and cell banks prepared at passage 13: for experiments the cells were used between passage 13 and 23. To make ChiPS4-pmGFP, the cells were transfected with the PiggyBac and transposase constructs at passage 17 and GFP+ ve cells selected by fluorescence activated cell sorting at passage 19. The polyclonal cell line was banked at passage 23; for experiments the cells were used between passage 23 and 33. After two passages all cell lines were tested to check for the uniform expression of pluripotency markers (Oct4, Sox2, Nanog, SSEA-3, SSEA-4, TRA-1–60, and TRA-1–81) and absence of differentiation markers (SSEA-1, HNF-3 beta, beta-III-tubulin, and smooth muscle alpha-actinin) by immunofluorescence.

## hESC and hiPSCs culture and differentiation

hESC (H9 or DCX$^{Cit/Y}$ H1 line) and hiPSCs (ChIPS4) were maintained as feeder-free cultures in DEF-based medium supplemented with bFGF (30 ng/mL, Peprotech, Cat. No. 100-18B) and Noggin (10 ng/ml, Peprotech, Cat. No. 120–10C) on Geltrex matrix coated plates (10 µg/cm$^2$, Life Technologies, Cat. No. A1413302), and enzymatically passaged using TryPLselect (Thermofisher, Cat. No. 12563011). For PSC passaging, the medium was complemented by addition of the Rho kinase inhibitor Y-27632 (10 mM, Tocris, Cat. No. 1254).

For differentiation assays, PSC were plated on Geltrex matrix at 40,000 cells/cm$^2$ and shifted to N2B27 medium after 24 hr. Cells were then differentiated for 3 days (D) in N2B27 supplemented with 3 µM Chiron99021 (Tocris, Cat. No. 4953) and 20 ng/ml bFGF (PeproTech, Cat. No. 100-18B), in the presence of 50 ng/ml Noggin (Peprotech, Cat. No. 120–10C) and 10 ng/ml SB431542 (Tocris, Cat. No. 1614) from D2 to D4 to generate NMP-like cells, see *Verrier et al., 2018*. For further differentiation, NMP-like cells were passaged using PBS-EDTA 0.5 mM and seeded back at controlled density (10$^5$ cells/cm$^2$) on Geltrex (20 µg/cm$^2$, Life Technologies, Cat. No. A1413302) in the presence of Y-27632 (10 mM, Tocris, Cat. No. 1254). Cells were then cultured in N2B27 containing 100 nM all-trans-RA (SIGMA, Cat. No. R2625) for the indicated time to obtain and differentiate neural progenitors. First human neural rosettes were observed from D06 with this regime. All work with hESCs was undertaken in approval of the UK Stem Cell Bank steering committee (license number SCSC14-29).

## Immunofluorescence

Human embryonic tissue was provided by the Human Developmental Biology Resource (HDBR; https://www.hdbr.org) following fixation in 4% paraformaldehyde (PFA) for 2 hr at 4°C and chicken embryonic neural tube at indicated stages was preserved in the same way, prior to processing for cryo-sectioning using standard procedures. hESCs or iPSCs derived neural rosettes were fixed in 1% PFA overnight at 4°C, permeabilised with 0.1% Triton X-100 in PBS. Tissue sections or rosettes grown in ibdi dishes (μ-Slide eight well, Cat. No. 80826) in vitro were subject to a standard immunofluorescence protocol using commercial primary and secondary antibodies at concentrations indicated in Key resources table. Finally, sections were stained with DAPI to visualize cell nuclei. Fluorescent images were captured using a Leica SP8 confocal using a 20X air or 63X NA1.2 APO water immersion objectives. Images of protein localisation in human embryonic spinal cord have been deposited in the HDBR atlas https://hdbratlas.org/organ-systems/nervous-system/cns/spinal-cord/storeylab.html.

## Pm-eGFP plasmid construction for engineering hiPS cells

A plasmid was constructed for engineering hIPSCs with the piggybac transposase system using the aPX1 transposon plasmid (Timothy Sanders, University of Chicago), containing the cytomegalovirus (CMV) enhancer, chicken b-actin (CAG) promoter expression and including the Invitrogen Gateway RfA cassette, allowing for phiC31-mediated recombination from Gateway Entry vectors. Finally, the CAG promoter and Gateway cassette (cargo) are flanked by two Inverted Terminal Repeat sequences (ITR, TTAA) allowing the piggyback transposon action. To label the cell membrane, a plasma membrane (pm) associated palmitoylated fluorescent protein was generated by the addition of the 20-amino acid sequence of ratGAP-43 MLCCMRRTKQVEKNDEDQKI to the N-terminus of the monomeric enhanced GFP (eGFP) (K. Svoboda, Addgene plasmid 18696) through sequential PCR amplification to make a pm-eGFP sequence. This pm-eGFP sequence was then inserted into the aPX1 donor vector using the Gateway System (ThermoFisher).

## HiPSCs transfection and creation of ChIPS4- Pm-GFP line

HiPSCs (Cellartis hIPS4 cell line) cells were transfected using the Neon transfection System (Life Technologies, Cat. No. MPK10025). Briefly, a mix of aPX1-pm-eGFP construct and piggyback transposase (*Yusa et al., 2011*), with a final concentration of 1–2 μg/μl, was added to a solution of $10^7$ cells and a single pulse of 1100 V with 30 ms duration was delivered via the Neon transfection system. Transfected cells were then plated back on Geltrex coated dishes (10 μg/cm$^2$, Life Technologies) and cultured in DEF-based medium supplemented with bFGF (30 ng/ml, Peprotech) and Noggin (10 ng/ml, Peprotech). In this way, a polyclonal line ChIPS4-pm-GFP was created, and this was used in the experiments reported here.

## Chick embryo transplantation assays

### Isochronic transplantation assay

Fertilized chicken eggs (*Gallus gallus domesticus*) from a commercial source (Winter Farm, Hertfordshire-Royston, UK) were incubated at 38°C to Hamburger and Hamilton (HH) stage 11–12 (13–16 somites) (*Hamburger and Hamilton, 1951*). ChIPS4-pm-eGFP were differentiated to NMP-Like cells and these were then replated and for cultured 6 days in neural differentiation conditions (as described above), and newly formed human neural rosettes were then manually micro-dissected using a sharp glass needle (0.290–0.310 mm) (Vitrolife). After resting in $CO_2$ incubator at 37°C for 2 hr, cell aggregates (diameter ≈ 50–100 μm) were gently dissociated and implanted into the dorsal region of one side of the neural tube of stage HH 11–12 chicken embryos. For transplantation of small groups of human dorsal neural progenitors, micro-dissected single human rosettes were further sub-dissected and mechanically dissociated prior to transplantation. Before implantation, the dorsal region on one side of the neural tube was removed at the level of the most recently formed somite (Figure S3). Embryos were then harvested 2 or 5 days post-transplantation (Stage HH 23–24 and HH 30–31 respectively). Chicken embryos were fixed in 4% PFA overnight at 4 °C, rinsed in PBS and left in PBS/30% sucrose solution overnight at 4 °C, then embedded and in agar, frozen and cryo-sectioned at 20 μm and subjected to immunofluorescence processing (as above).

### Heterochronic transplantation assay

To create an asynchrony between the donor (human neural progenitors) and the host (chicken embryos), newly formed human neural rosettes were micro-dissected after 6 days of neural differentiation from

NMP-L cells, gently dissociated and implanted into the dorsal region of one side of the neural tube of stage HH 20–21 chicken embryos. Before transfer, the dorsal region on one side of the neural tube was removed at the mid-trunk, thoracic region (just posterior to the forelimb region). Similar to the isochronic transplantation assay, embryos were then harvested 2 or 5 days post-transplantation (stage HH 26–27 and HH 32–33, respectively). Chicken embryos were then fixed, processed and subjected to immunofluorescence as described above.

## Isotopic and isochronic transplantation of GFP chicken neural tube

Fertilized chicken eggs (*G. gallus domesticus*) from a commercial source (Winter Farm, Hertfordshire-Royston, UK) (wild type) or from a transgenic chicken line expressing a myristoylation sequence fused to GFP (myr-GFP) which localises to the plasma membrane (National Avian Research Facility, Roslin Institute, see *Rozbicki et al., 2015*), were incubated at 38°C to stage HH11–12 (13–16 somites). Briefly, the neural tube of the myr-GFP chicken embryos was removed and then micro-dissected to keep only the dorsal region of one side. Before transplantation, the dorsal region on one side of the host wild-type neural tube was removed at the rostro-caudal level of the most recently formed somite. Embryos were harvested 2 days post-transplantation (stage HH23–24). Chicken embryos were then fixed in in 4% PFA, processed and subjected to immunofluorescence as described above.

## Cell counts and statistical analysis

At D4 of in vitro differentiation from NMP-like cells, defined fields (200 µm$^2$) of neural progenitors forming a neural plate-like shape were selected for quantification. At later time points, human neural rosettes were selected for analysis and quantification by their shape (round with a single lumen) and size (neural rosette diameter, D6: 70–100 µm, average: 84±2.7 µm; D10: 125–137 µm, average: 129±1.5 µm; D14: 155–195 µm, average: 179 ±2.7 µm, *Figure 1—source data 3*). The proportion of cells expressing marker proteins of interest was determined by counting positive cells within a field of neural progenitors (D4) or in three optical sections in a rosette (D6 and later, includes rosette and contacting cells at its periphery at later timepoints) out of total (DAPI+ or GFP+) cells. Fifteen fields of neural progenitors (D4) or 15 rosettes (D6 and later) were analysed for each marker (*n* = 15, five fields of neural progenitors or rosettes from each of three independent differentiation experiments). For transplanted human rosettes at least three sections from grafts in three different embryos were used to determine the proportion of cells expressing a protein of interest. Transplanted human cells were only counted when cells were positive for GFP, immunoreactivity for protein of interest and DAPI. To avoid counting the same cells in consecutive sections, alternate sections were used for each different immunolabelling assay (four different immunofluorescence assays/transplanted embryo) (Figures source data). Statistical analysis was performed using non-parametric Mann-Whitney U test for non normally distributed data using Prism V8. Results are represented as means ± SEM or SD as indicated in figures and p-values are indicated with *p<0.05, **p<0.01, *** p<0.001, and ****p<0.0001.

## Acknowledgements

We are grateful to Aida Rodrigo-Albors and to Teresa Rayon and James Briscoe for critical reading of versions of this manuscript. We thank Timothy Sanders, University of Chicago for the aPX1 transposon plasmid, Dr Boaz Levi (Allen Institute) and WiCell for the hESC H1 line expressing DCX$^{Cit/Y}$ and WiCell for the hESC line H9. All work with hESCs was undertaken in approval of the UK Stem Cell Bank steering committee (license number SCSC14-29). We acknowledge The National Avian Research Facility at The Roslin Institute, University of Edinburgh and Kees Weijer, University of Dundee, for myr-GFP transgenic chicken embryos. Human embryonic tissue (4–6 weeks of gestation) was obtained from the MRC/ Wellcome-Trust (grant no. 006237/1) funded Human Developmental Biology Resource (HDBR; https://www.hdbr.org) with appropriate maternal written consent and approval from the London Fulham Research Ethics Committee (18/LO/ 0822) and the Newcastle and North Tyneside NHS Health Authority Joint Ethics Committee (08 /H0906/21 + 5). HDBR is regulated by the UK Human Tissue Authority (HTA; https://www.hta.gov.uk) and operates in accordance with the relevant HTA codes of practice. This work was part of project no. 200,407 registered with the HDBR. This research was supported by a Wellcome Investigator award WT102817AIA and an enhancement award WT102817/Z/13 /A to KGS. The confocal microscope used for imaging was purchased with support of a Wellcome Trust Multi-User Equipment grant (WT101468).

## Additional information

### Funding

| Funder | Grant reference number | Author |
| --- | --- | --- |
| Wellcome Trust | WT102817AIA | Kate G Storey |
| Wellcome Trust | WT102817/Z/13/A | Kate G Storey |
| Wellcome Trust | WT101468 | Kate G Storey |

The funders had no role in study design, data collection and interpretation, or the decision to submit the work for publication.

### Author contributions

Alwyn Dady, Conceptualization, Data curation, Formal analysis, Investigation, Methodology, Visualization, Writing - original draft, Writing - review and editing; Lindsay Davidson, Investigation, Methodology, Writing - review and editing; Pamela A Halley, Investigation, Methodology; Kate G Storey, Conceptualization, Funding acquisition, Methodology, Project administration, Resources, Supervision, Writing - original draft, Writing - review and editing

### Author ORCIDs

Alwyn Dady (iD) http://orcid.org/0000-0002-5659-8801
Lindsay Davidson (iD) http://orcid.org/0000-0002-2927-4377
Pamela A Halley (iD) http://orcid.org/0000-0002-6970-7835
Kate G Storey (iD) http://orcid.org/0000-0003-3506-1287

### Ethics

Human subjects: Human embryonic tissue (4 to 6 weeks of gestation) was obtained from the MRC/Wellcome-Trust (grant no. 006237/1) funded Human Developmental Biology Resource (HDBR; https://www.hdbr.org) with appropriate maternal written consent and approval from the London Fulham Research Ethics Committee (18/LO/ 0822) and the Newcastle and North Tyneside NHS Health Authority Joint Ethics Committee (08/H0906/21+5). HDBR is regulated by the UK Human Tissue Authority (HTA; www.hta.gov.uk) and operates in accordance with the relevant HTA codes of practice. This work was part of project no. 200407 registered with the HDBR. Human ESC lines H9 and H1 expressing DCXCit/Y were provided by WiCel and all work with hESCs was undertaken in approval of the UK Stem Cell Bank steering committee (license number SCSC14-29).

### Decision letter and Author response

Decision letter https://doi.org/10.7554/eLife.67283.sa1
Author response https://doi.org/10.7554/eLife.67283.sa2

## Additional files

### Supplementary files

• Transparent reporting form

### Data availability

All data generated or analysed during this study are included in the manuscript and supporting files. Source data (all numerical meta data) are provided as excel tables aligned to the relevant figure.

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
