## [Editor Report]

This manuscript is of interest to a large audience in the fields of stem cells, developmental biology and neural regeneration. The authors assess the roles of extrinsic versus intrinsic signalling on differentiation of human neural cells by comparing their differentiation rates across different environments (in vitro, in the human embryo and grafted into a chicken embryo).

---

## [Decision Letter]

**Decision letter after peer review:**

Thank you for submitting your article "Rapid pace of human spinal cord differentiation in vitro is only initially maintained in a heterologous environment" for consideration by *eLife*. Your article has been reviewed by 3 peer reviewers, and the evaluation has been overseen by Marianne Bronner as the Senior Editor and Reviewing Editor. The following individual involved in review of your submission has agreed to reveal their identity: Tatiana Solovieva (Reviewer #3).

The pace that determines differentiation of embryonic cells (and ESCs/iPS) in general, and human embryonic cells in particular, is at the forefront of stem cells and developmental biology fields, with promising translational perspectives towards regenerative medicine. In this manuscript Dady et al. used human neural rosettes derived from hESCS/iPS to test their differentiation pace in the dish. Their data elegantly show their sequential differentiation into different types of neurons and glia at distinct time windows. When compared to their corresponding cells/domain within human embryonic spinal cords, the authors discover that, although the in vitro rosettes recapitulate the temporal sequence of neural tube differentiation (but into dorsal neural subpopulations only), they proceed much more rapidly in vitro compared to the intact embryo. This analysis led the authors to conclude that spinal cord progenitors in the in vitro conditions accelerate their developmental pace as compared to the embryonic counterparts due to lack of the endogenous environmental context.

Intriguingly, when the rosettes were grafted into chick neural tube in ovo, they did not adjust to the new environment, which contains much more rapidly differentiating cells, and remained in their initial pace. Later, the grafted cells halted their development and remained progenitors. These findings led the authors to conclude that intrinsic properties of the neural rosettes, rather than extrinsic signals from the avian spinal cord, govern their differentiation pace.

Essential revisions:

All of the points raised by the reviewers seem both reasonable and feasible. Most importantly:

1. Heterochronic transplantation is a key experiment to test the notion of intrinsic regulation of the timing of human neural stem cell differentiation.

2. Please provide information on whether the cells analyzed for a given marker are present within or outside of that marker domain in the chick host. This is crucial to interpretation of the data.

3. Better analysis of the stalled grafts in terms of their cell death/proliferation /glia markers.

4. As a control, chick-to-chick graft of early progenitors to compare with the human graft to rule out the possibility of an experimental artifact that leads to the human cell behaviour in the chick.

5. A few grafts using smaller groups of rosette derived cells to clarify whether or not the current results represent human cells that are receiving largely chick host derived signals.

*Reviewer #1 (Recommendations for the authors):*

In this interesting paper by Dady and colleagues the nature of human neural progenitor differentiation is evaluated via transplantation studies. The first part of the paper establishes the timing of neural differentiation in human IPSC model systems and in human embryonic spinal cord, showing that the relative timing of neurogenesis and gliogenesis is maintained. In the second part of the paper these human IPSC neural rosettes are transplanted into the chick spinal cord during neurogenic stages (i.e. Isochronic transplantation) and they find that neurons are generated by these transplanted populations. Analysis of transplants at later stages reveals that neurogenesis has "stalled" and is relatively reduced within the transplanted population.

Overall, this is an interesting paper that uses classic approaches to answer potentially interesting questions, however there are some issues that limit its potential impact. The first two figures are recursive and show that the authors can implement an existing protocol. The transplantation studies are intriguing by fall short of offering any new insights. The key finding seems to be that at later stages post-transplantation neuronal differential is "stalled". There are many other reasons (besides stalling") that could explain their results. Suppose that stalling was indeed occurring, the authors offer no cellular or molecular insights into what regulates these intrinsic differences across species. At the end, I'm left asking what we have really learned about the mechanisms that control the timing (pace is just fancy and new way of saying timing) of differentiation in human neural stem cells. Below are a few experimental suggestions that could add more insight into this paper:

1) Fig7, late stage transplant, should check NFIA expression. E7 is gliogenic in the chick and its possible that the rosettes are now becoming gliogenic. Perhaps the observed "stalling" is really a switch to gliogenesis. Alternatively, the apparent reduction in the proportion of neurons could be a survival issue, where older and more differentiated neurons from human rosettes cannot survive in the chick spinal cord. Thus, cell death should be evaluated. Even if there are no detectable changes in cell death, the interpretation should be dialed back a bit as its totally possible that there issues with the maintenance of human neurons in chick spinal cord and by the time they are analyzing transplants at E7 in chick the rosettes have switched to gliogenesis and are no longer making neurons (and the neurons that were produced have died off).

2) The phrasing is a bit odd. Homo-chronic should be phrased as "Isochronic". "Pace" is really just timing of differentiation.

3) To truly test whether there is an intrinsic mechanism at play in these transplantation studies, they should perform the heterochronic experiment: Transplant Neurogenic D6 rosettes into the E6 gliogenic chick spinal cord. Admittedly, this is a tough experiment. Therefore, as an alternative, they should perform the complementary heterochronic transplantation: Gliogenic D20 rosettes into the E2, neurogenic chick spinal cord. If there is some sort of intrinsic timing mechanism at play, then the D20 rosettes should still make glia in the neurogenic environment. This was actually already shown for mouse glial progenitors (Mukouyama, et al. PNAS 2006) and it would be a nice addition to this paper.

4) The labeling in figure 4b makes no sense and seems arbitrarily defined. First, a better description of these regions and how they are defined needs to be included. Second, a representative cell or image from these regions should be shown at high magnification. As it stands, I'm having a very hard time deciphering what is different about these groups. Third, each image that contains this type of analysis (for example 4d) should show the respective regions.

5) Figure 3 is really useful and interesting. Would be good to show the human staining alongside the chick (or mouse) staining. The diagram in 3I is nice, but side-by-side images across species would really make it more compelling.

*Reviewer #3 (Recommendations for the authors):*

1. It would be helpful to see individual channels for the markers visualised in the chick cross sections in Figure 4D and E. It appears as though the intensity of SOX2 and PAX7 in GFP+ cells is reduced relative to expression in the neighbouring chick neural tube cells, but it is difficult to tell without seeing the SOX2 and PAX7 channels separately. When counting cells positive or negative for a marker gene, is intensity of that marker within a cell factored in?

---

## [Author Response]

Essential revisions:All of the points raised by the reviewers seem both reasonable and feasible. Most importantly:1. Heterochronic transplantation is a key experiment to test the notion of intrinsic regulation of the timing of human neural stem cell differentiation.

Heterochronic transplantation of D6 human neural rosettes into the same position in the developing spinal cord but into older embryos has been undertaken. Three independent experiments were carried out, with three sections from 3 grafts per experiment analysed for each of our panel of markers at 2 days and at 5 days post-transplantation. These extensive new data are presented in a new Results section and in new figures 8 and 9, and results summary figure 10. We find that as with isochronic transplantation there is a >2-day delay before human cells respond to the chick environment, in both contexts the rosette cells retain differentiation timing found in rosettes cultured in vitro. This supports the conclusion that human cells initially follow cell instinct differentiation timing. Strikingly, by 5 days heterochronic grafts exhibit increased neurogenesis in comparison with in vitro rosettes, (while isochronic grafts showed reduced neurogenesis). This demonstrates that human cells can respond to differentiation signals in the chicken embryonic environment. Moreover, by heterochronic day 5 the chicken host has already embarked on gliogenesis, but the human cells continue to follow their normal neural differentiation programme, entering the neurogenic phase. As human cells do not obviate neurogenesis in a gliogenic environment these data further show that human cells follow an intrinsic differentiation programme.

­­­

2. Please provide information on whether the cells analyzed for a given marker are present within or outside of that marker domain in the chick host. This is crucial to interpretation of the data.

This is only an issue for one marker we use (ISLET1) which has a discrete localisation in the dorsoventral axis. The number of sections with GFP +ve human cells/grafts in this precise location in the host embryo is now included in the Results/ figure legends. We have provided further information about this in our detailed Response to reviewer (3) comments. While not all sections through grafted rosettes include such cells, we have examples from 3 (isochronic) or 4 (isochronic small cell group) grafts/embryos.

3. Better analysis of the stalled grafts in terms of their cell death/proliferation /glia markers.

This has been addressed by assessing expression of key marker proteins in sections of isochronic day 5 grafts. Analysis of cell death using an antibody to detect activated caspase 3 revealed one or two positive cells in only a few sections. Cell proliferation detected with an antibody that recognises Ki67 showed that almost all human rosette cells expressed this protein. These data are quantified and provided in revised figure 6 and as metadata. We conclude that cell death does not contribute to the differentiation state observed at Day 5, although we cannot formally exclude the possibility that this takes place discretely between day 2 and day 5 and so was missed in this analysis (noted in the manuscript). Our conclusion is supported by the proliferative status (Ki67) of almost all cells, which further confirms our original analysis showing that most cells express SOX2 and PAX7. We did not have enough sections to also assess the possibility that these cells have been induced into glial progenitors, but the extensive expression of the neurogenic neural progenitor marker PAX7 in these grafts supports the conclusion that this could not account for the apparent stalling of the neurogenic differentiation. Overall, these data strongly suggest that iso-chronically grafted human neural progenitors have been exposed to insufficient levels of differentiation signals to progress the neural differentiation programme. This is consistent with the slower cell cycle of human neural progenitors and slower metabolism of human cells delaying response to the signalling environment in comparison with the cells of the host chicken embryo. This conclusion is further supported by the increased neurogenesis detected in day 5 hetero-chronic grafts, where exposure to a more developmentally advanced environment now drives neurogenesis.

4. As a control, chick-to-chick graft of early progenitors to compare with the human graft to rule out the possibility of an experimental artifact that leads to the human cell behaviour in the chick.

This further control experiment has been undertaken using grafts of the dorsal neural tube from the myr-GFP transgenic chicken line and is presented in figure 4 supplemental figure S3. We observe normal patterns of contribution and expression of key protein markers in transplanted embryos. This confirms that transplantation operation/injury does not in itself lead to altered differentiation.

5. A few grafts using smaller groups of rosette derived cells to clarify whether or not the current results represent human cells that are receiving largely chick host derived signals.

These experiments have been carried out and are presented and quantified in Figure 4 Supplementary figure and as metadata. These revealed essentially the same results as previous experiments. They control for the possibility that the single cells and small cell groups in our initial analysis may have become separated at a later time and so did not represent cells that had been fully integrated into the chicken neural tube throughout the time leading up to their analysis.

Reviewer #1 (Recommendations for the authors):In this interesting paper by Dady and colleagues the nature of human neural progenitor differentiation is evaluated via transplantation studies. The first part of the paper establishes the timing of neural differentiation in human IPSC model systems and in human embryonic spinal cord, showing that the relative timing of neurogenesis and gliogenesis is maintained. In the second part of the paper these human IPSC neural rosettes are transplanted into the chick spinal cord during neurogenic stages (i.e. Isochronic transplantation) and they find that neurons are generated by these transplanted populations. Analysis of transplants at later stages reveals that neurogenesis has "stalled" and is relatively reduced within the transplanted population.Overall, this is an interesting paper that uses classic approaches to answer potentially interesting questions, however there are some issues that limit its potential impact. The first two figures are recursive and show that the authors can implement an existing protocol.

Our previous work established this differentiation protocol (Verrier et al. 2018 Development), but this is its first use to analyse by immunofluorescence the timing of neural differentiation and the appearance of specific neuronal and glial cell types in an in vitro neural rosette assay. None of the data presented is recursive and it provides a quantitative timeline of human dorsal spinal cord differentiation. Moreover, our comparison with the human embryonic spinal cord indicates that neural differentiation progresses more rapidly in vitro.

The transplantation studies are intriguing by fall short of offering any new insights. The key finding seems to be that at later stages post-transplantation neuronal differential is "stalled". There are many other reasons (besides stalling") that could explain their results. Suppose that stalling was indeed occurring, the authors offer no cellular or molecular insights into what regulates these intrinsic differences across species. At the end, I'm left asking what we have really learned about the mechanisms that control the timing (pace is just fancy and new way of saying timing) of differentiation in human neural stem cells. Below are a few experimental suggestions that could add more insight into this paper:1) Fig7, late stage transplant, should check NFIA expression. E7 is gliogenic in the chick and its possible that the rosettes are now becoming gliogenic. Perhaps the observed "stalling" is really a switch to gliogenesis.

We appreciate that the E7 chick spinal cord is gliogenic (Deneen et al. 2006) and it is formally possible that the human rosettes start to become gliogenic in this context at an earlier time point than in vitro. We observe in vitro that the human rosettes become gliogenic quite late, at D20, while analysis of grafted D6 rosettes after 5 days takes us only to D11 in vitro when rosettes are clearly neurogenic (Figure 2), so the chick environment would need to be able to dramatically accelerate or divert the human neural differentiation programme. Unfortunately, this is the only experiment we have been unable to address directly as we did not have enough sections of day 5 iso-chronically grafted rosettes to assess NFIA expression. However, below we set out the evidence for why we think these late-stage iso-chronically transplanted rosettes are not gliogenic.

We have effectively tested this question in a new hetero-chronic grafting experiment (undertaken in response to your further comments below), which assesses differentiation of D6 human rosettes grafted into an E3 older chick spinal cord; after 2 days we did not detect expression of glial marker (NFIA) in transplanted human rosettes, despite NFIA expression in the host chicken embryo (new Figure 8D). Further, after 5 days, we found a large increase in the number of neurons (compared to in vitro rosettes), excluding an early switch to gliogenesis at this time (new Figure 9). These findings indicate that the transplanted human rosette cells can respond to the chicken host environment after this longer period of exposure to a more differentiated environment, indeed this is consistent with continued neurogenesis during the transition to gliogenesis that is observed in the developing chick spinal cord, with neurons still being generated at E6 (Deneen et al. 2006). As expanded upon below, these findings support the interpretation that D6 rosettes proceed along the normal sequence of the human neural differentiation programme, generating neurons in a supportive environment, rather than being reprogrammed to the gliogenic pathway. D6 rosettes grafted iso-chronically into the earlier E2 embryo will be exposed to a less advanced neurogenic environment and we think this is not enough to sustain productive human neurogenesis, but some P27 and ISLET1 positive cells are generated from rosette cells that continue to express neurogenic progenitor protein PAX7 at day 5 (Figure 6D) (this is normally downregulated by E6 in the chick during gliogenesis transition, Deneen et al. 2006) and so it is unlikely that the rosette cells are gliogenic. We have now summarised the differentiation status of rosette cells in these different contexts in a new schematic figure 10.

Alternatively, the apparent reduction in the proportion of neurons could be a survival issue, where older and more differentiated neurons from human rosettes cannot survive in the chick spinal cord. Thus, cell death should be evaluated. Even if there are no detectable changes in cell death, the interpretation should be dialed back a bit as its totally possible that there issues with the maintenance of human neurons in chick spinal cord and by the time they are analyzing transplants at E7 in chick the rosettes have switched to gliogenesis and are no longer making neurons (and the neurons that were produced have died off).

We have evaluated cell death in day 5 iso-chronically grafted rosettes and found only a few cells expressing cleaved casapase3 (new Figure 6H). We also analysed cell proliferation and observed that most transplanted human rosette cells express Ki67 indicative of cycling cells (new Figure 6G). These data, along with the expression of SOX2 and PAX7 in almost all cells in iso-chronically transplanted human rosettes after 5 days (Figures 6C, D) indicate that these retain a proliferative neural progenitor cell state, while P27 and Islet1 cell counts indicate they produce few neurons. Moreover, as noted above the same D6 rosettes grafted into a later E3 embryo generate increased numbers of neurons (greater than in vitro) by day 5, indicating that such human cells survive well in the chicken environment. However, it is formally possible that neuronal death takes place discretely between day 2 and day 5 in iso-chronically grafted rosettes and so was not detected in this analysis. We now note this in the Discussion.

2) The phrasing is a bit odd. Homo-chronic should be phrased as "Isochronic". "Pace" is really just timing of differentiation.

We used the word “homo-chronic” for a reason. The term iso-chronic is often used in classical experiments in which the same tissue is transplanted between two embryos at exactly the same stage of development for example to make fate maps (e.g., Catala et al.1996 MOD, PMID: 7669693). As we did not transplant a piece of human neural tube for evident ethical reasons, we found the term “homo-chronic” more relevant as we are comparing two similar tissues at the same stage of differentiation from two different species. However, for clarity we have now replaced the term homo-chronic with by iso-chronic. We also appreciate that timing and pace have similar meanings here.

3) To truly test whether there is an intrinsic mechanism at play in these transplantation studies, they should perform the heterochronic experiment: Transplant Neurogenic D6 rosettes into the E6 gliogenic chick spinal cord. Admittedly, this is a tough experiment. Therefore, as an alternative, they should perform the complementary heterochronic transplantation: Gliogenic D20 rosettes into the E2, neurogenic chick spinal cord. If there is some sort of intrinsic timing mechanism at play, then the D20 rosettes should still make glia in the neurogenic environment. This was actually already shown for mouse glial progenitors (Mukouyama, et al. PNAS 2006) and it would be a nice addition to this paper.

We have now carried out a heterochronic experiment. We opted to graft D6 human rosettes into an older host chick embryo at E3, as this allowed us to compare data with D6 rosettes cultured in vitro or iso-chronically transplanted, after 2 days and 5 days. Grafting into even older E6 embryos, as suggested here, was not possible because these are highly vascularised with a strong heartbeat that makes the transplantation and positioning of rosettes technically challenging and reduces embryo viability. The alternative suggestion, transplanting gliogenic D20 rosettes was not undertaken because by this stage the rosette structure has been outgrown and we would also have to characterise rosettes in vitro for a further 2 (D22) and 5 (D27) days to provide in vitro control data sets (note each analysis is carried out with 3 independent differentiations from hPSCs and cells in >5 rosettes quantified and analysed for each marker assessed).

So, as presented in detail in the paper, we transplanted D6 rosettes composed of dorsal neural progenitors into the dorsal region of the neural tube of an older E3 chicken embryo. After 2 days the E3 host embryo had reached E5 and had begun the gliogenic switch, expressing NFIA in the dorsal neural tube (new Figure 8D). Importantly, we did not detect NFIA in the transplanted rosette after 2 days. In fact, we found no difference between hetero-chronically or iso-chronically transplanted and in vitro cultured rosettes at day 2, indicating that in this timeframe transplanted human cells were not influenced by the chicken host environment. We argue in the paper that this finding is consistent with the correlation between greater protein stability / slower human cell cycle time and slower differentiation pace (Rayon et al. 2020) and we make a case that transit through the cell cycle may be required for expression of differentiation genes and that this may explain the delay seen in both conditions in the response of human cells to the chicken environment.

Importantly, 5 days after heterochronic grafting (when the human cells had been exposed to the gliogenic environment for 3 days), we found increased neuron production (both p27 and islet1 +ve cells) compared with in vitro rosettes and iso-chronic transplantation conditions (new Figure 9). This tells us that by day 5 human cells can now respond to the chicken environment, and while in isochronic this is insufficient to drive neurogenesis, in the hetero-chronic conditions this results in the next step the neural differentiation programme, i.e. neuron production.

These heterochronic grafting data therefore support the conclusion that the human neural differentiation programme progresses in an environment with an appropriate level of differentiation signals (presented over 5 days), and that the sequence of cell types generated by this programme is not altered in this older and newly gliogenic environment. This is consistent with the findings of Mukouyama, et al. in the mouse, in that they find that Olig2+ progenitors progress through a differentiation programme, first neurogenic phase and then gliogenic phase when transplanted into the chicken environment, and that this is not reversible, older gliogenic Olig2+ progenitors cannot be reprogrammed back to a neurogenic phase in younger chicken host. Here we show that the neurogenic phase of differentiating human neural progenitors is not obviated by development in an increasingly gliogenic environment. We now address this in the Discussion, noting that this supports operation of an intrinsic programme of human neural differentiation.

4) The labeling in figure 4b makes no sense and seems arbitrarily defined. First, a better description of these regions and how they are defined needs to be included. Second, a representative cell or image from these regions should be shown at high magnification. As it stands, I'm having a very hard time deciphering what is different about these groups. Third, each image that contains this type of analysis (for example 4d) should show the respective regions.

We thank the reviewer for noting this and have clarified this in the text, opting to replace the numbers with abbreviations to remind readers which cell configuration is indicated (Figure 4 and second para page 11). Please note that examples of all three cell configuration types scored, (cells in a rosette formation abutting/continuous with the chicken neural tube (contiguous rosette, CR), directly incorporated into the chick neuroepithelium (incorporated cell group, ICG), or single cells incorporated within the chicken neuroepithelium (incorporated single cells, ISC)), do not appear in all sections – so for example, while all 3 are apparent in Figures 4C and 4E, only two contiguous rosette and incorporated cell groups ( i.e. no single cells here) appear in Figure 4D.

5) Figure 3 is really useful and interesting. Would be good to show the human staining alongside the chick (or mouse) staining. The diagram in 3I is nice, but side-by-side images across species would really make it more compelling.

While we agree this would be nice, we do not have precisely the same immunofluorescence images with these different species to include in this figure. The point of the figure is to compare differentiation of human neural rosettes in vitro with that in the human embryonic spinal cord.

Reviewer #3 (Recommendations for the authors):1. It would be helpful to see individual channels for the markers visualised in the chick cross sections in Figure 4D and E. It appears as though the intensity of SOX2 and PAX7 in GFP+ cells is reduced relative to expression in the neighbouring chick neural tube cells, but it is difficult to tell without seeing the SOX2 and PAX7 channels separately. When counting cells positive or negative for a marker gene, is intensity of that marker within a cell factored in?

Separate channels for marker proteins and GFP are now provided in revised figure 4. We have noticed that PAX7 expression appears lower in human compared to chick neural progenitors, however, we do not know whether this reflects different levels of protein in different species or different detection sensitivities with this antibody, so we have not drawn any conclusions from this observation. The way we scored the positive cells for each marker is described in the material and methods section. We do not factor in the intensity of expression of the different markers. We scored cells as positive when human cells were clearly positive for GFP, a marker and DAPI.